# Conformal Uncertainty Indicator for Continual Test-Time Adaptation

## Abstract

Continual Test-Time Adaptation (CTTA) enables models to adapt to sequential domain shifts during testing, but reliance on pseudo-labels makes them prone to error accumulation. Reliable uncertainty estimation is thus critical. We study this problem under the calibration-aided CTTA setting, where a small calibration buffer from the source domain is available as reference. We propose the Conformal Uncertainty Indicator (CUI), a plug-and-play method that leverages Conformal Prediction (CP) with calibration data. Unlike standard CP, which suffers from a coverage gap under domain shifts, CUI jointly measures model shift and data shift to adjust conformal quantiles and restore coverage. The resulting prediction set size provides a reliable indicator of test-time uncertainty. Building on this, we introduce a CUI-guided adaptation strategy that updates models only on confident samples. Experiments on three benchmarks show that CUI achieves accurate uncertainty estimation and improves the robustness of multiple CTTA baselines.

## 1 Introduction

Recently, Continual Test-Time Adaptation (CTTA) (Wang et al., 2022) has attracted significant attention for enabling trained models to handle sequential domain shifts through self-adaptation. However, in many high-stakes scenarios, *the cost of incorrect predictions is prohibitively high*, as in autonomous driving (Sójka et al., 2023) and medical imaging (Chen et al., 2024), where even a single error can cause serious risks and errors may accumulate during continual adaptation. To address this issue, it is crucial to evaluate the reliability of each prediction before using it for adaptation. Uncertainty estimation provides a common approach, but existing methods such as Bayesian approximation (Maddox et al., 2019), Monte Carlo dropout (Gal & Ghahramani, 2016), or entropy-based scores (Shi et al., 2024) are either computationally expensive or prone to overconfidence, and thus less effective for continual adaptation.

This limitation is further exacerbated by the strict CTTA setting, which assumes no source data are available and leaves prediction evaluation without any reliable reference. In many real-world scenarios, however, *it is acceptable and even necessary to maintain a small static calibration buffer of source samples to support reliable long-term adaptation*. We refer to this extended variant as *Calibration-aided CTTA (CCTTA)*, which forms the focus of this work. Our goal is to investigate how calibration data can be effectively exploited to enhance uncertainty estimation at test time.

The availability of calibration data in CCTTA naturally motivates the use of Conformal Prediction (CP) (Vovk et al., 2005), which provides a principled framework for uncertainty estimation. By constructing set-valued predictions, CP not only guarantees that the true label lies within the set with a pre-specified probability but also uses the set size (often referred to as inefficiency) as a natural measure of prediction uncertainty. It further offers two desirable properties: it is *model-agnostic*, requiring no assumptions or modifications to the underlying model, and it provides *controllable coverage*, ensuring that uncertainty estimates are statistically valid. These features make CP an attractive candidate for continual adaptation, where reliable uncertainty quantification is essential for avoiding error accumulation. However, applying CP in continual domain shift scenarios is far from straightforward. Classical CP relies on the assumption of data exchangeability, meaning that the order and distribution of observations are assumed not to change. This assumption is violated under distribution shift, leading to a **coverage gap** in which the actual coverage falls far below the nominal guarantee (Barber et al., 2023), and uncertainty estimates become untrustworthy in practice.

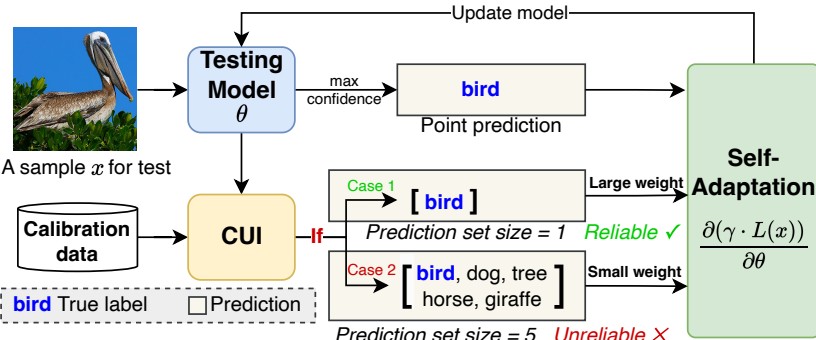

Figure 1: Illustration of the proposed CUI for calibration-aided CTTA. CUI leverages conformal prediction to produce set-valued outputs, where small sets indicate reliable samples for stronger adaptation and large sets reflect higher uncertainty. By compensating the coverage gap, CUI enables trustworthy uncertainty estimates to guide reliable adaptation.

To close the coverage gap, we propose Conformal Uncertainty Indicator (**CUI**), a **plug and play** uncertainty estimator for CCTTA. CUI uses a small labeled calibration set from the source domain and jointly considers model shift, reflected in how the model's predictions deviate from calibration behavior, and data shift, captured by representation discrepancies between calibration and test samples. These signals are used to adaptively correct the conformal quantile, thereby compensating for the violation of exchangeability and restoring reliable coverage. As a result, the size of the prediction set becomes a trustworthy indicator of test-time uncertainty. Furthermore, we introduce a CUI-guided adaptation strategy that updates the model only on confident samples, improving robustness of existing CTTA methods without additional supervision. Our contributions are three-fold:

(1) We propose the Conformal Uncertainty Indicator (CUI), a plug-and-play uncertainty estimator for CTTA that leverages a small calibration buffer from the source domain.

(2) CUI addresses the coverage gap of conformal prediction under domain shifts by jointly modeling model shift and data shift to calibrate prediction sets, making set size a reliable measure of test-time uncertainty.

(3) We further introduce a CUI-guided adaptation strategy that selectively updates models on confident samples, improving the robustness of multiple CTTA baselines across benchmark datasets.

## 2 RELATED WORK

**Continual Test-Time Adaptation**. Test-Time Adaptation (TTA) enables source-free, online adaptation of a model to target domain characteristics (Jain & Learned-Miller, 2011; Sun et al., 2020; Wang et al., 2020). CTTA (Wang et al., 2022) extends TTA to sequentially changing domains, addressing long-term adaptation but suffering from error accumulation and forgetting (Tarvainen & Valpola, 2017; Wang et al., 2022). Mean-teacher approaches (Tarvainen & Valpola, 2017) stabilize learning via exponential moving averages, while augmentation-averaged predictions (Wang et al., 2022; Brahma & Rai, 2023; Döbler et al., 2023; Yang et al., 2023) increase robustness to out-of-distribution inputs. Contrastive objectives (Döbler et al., 2023; Chakrabarty et al., 2023) preserve semantic consistency, and parameter restoration (Wang et al., 2022; Brahma & Rai, 2023) prevents forgetting. Most existing methods are developed under the strict CTTA setting, where no source data are available once deployment begins. Although this constraint enforces a fully source-free scenario, it also leaves prediction evaluation without any reliable reference information, making calibrated uncertainty estimation infeasible in practice.

**Conformal Prediction**. CP (Vovk et al., 2005) provides a principled framework for quantifying uncertainty by generating prediction sets that contain the true label with a user-specified probability. Its distribution-free validity and model-agnostic nature make it appealing for safety-critical applications, including medical diagnostics (Caruana et al., 2015), autonomous driving (Lekeufack et al., 2023), and finance. Recent work has extended CP to risk control and complex prediction scenarios (Farinhas et al., 2023). However, standard CP relies on the assumption of exchangeability, which breaks under domain shifts and leads to a coverage gap (Barber et al., 2023). To the best of our knowledge, conformal prediction has not been explored in CTTA, where estimating uncertainty in long-term, continually changing test environments is especially crucial.

## 3 PRELIMINARY: CTTA AND CP

### 3.1 CALIBRATION-AIDED CONTINUAL TEST-TIME ADAPTATION

CTTA methods adapt a pre-trained classification model from a source domain to unlabeled target streams $\mathcal{D}^k = \{x_m^k\}_{m=1}^{N^k}$, where $k$ indexes the target domain. At each step, the model must both provide a prediction and update itself without access to ground-truth labels. Existing approaches are typically studied under the strict CTTA setting, where no source data are accessible after deployment. While this enforces a fully source-free protocol, it leaves prediction evaluation without any reference information, making calibrated uncertainty estimation infeasible. In contrast, the Calibration-aided CTTA (CCTTA) setting allows a small static calibration buffer from the source domain, providing valuable reference data for estimating prediction reliability during continual adaptation. This paper focuses on uncertainty estimation under the CCTTA setting. To achieve this, we require a principled framework that can provide statistically valid uncertainty estimates, and conformal prediction (CP) offers a natural foundation.

### 3.2 CONFORMAL PREDICTION AND COVERAGE GAP ISSUE

We introduce CP under a classification task. Let $\mathcal{X}$ be the input space and $\mathcal{Y} := \{1, \cdots, K\}$ be the label space. We use $\pi : \mathcal{X} \to \mathbb{R}^K$ to denote the pre-trained model that is used to predict the label of a test sample. The model prediction in this classification task is generally made as

$$\hat{y} = \arg\max_{y \in \mathcal{Y}} \pi(y|x), \tag{1}$$

where $\pi(y|x)$ can be seen as the confidence of that $x$ being labeled to class $y$. Such point predictions, however, do not quantify predictive uncertainty. Conformal prediction (CP) (Vovk et al., 2005) provides a distribution-free framework to address this by constructing a prediction set $\mathcal{P}(x) \subseteq \mathcal{Y}$ that contains the true label with high probability. Specifically, CP guarantees **marginal coverage**:

$$\mathbb{P}(y \in \mathcal{P}(x)) \geq 1 - \alpha, \tag{2}$$

for a user-specified error level $\alpha \in (0, 1)$. For example, setting $\alpha = 0.1$ ensures that the constructed prediction set includes the true label at least $90\%$ of the time.

However, the coverage guarantee in Eq. (2) holds only under the assumption that calibration and test data are *exchangeable*, i.e., drawn from the same distribution (Vovk et al., 2005; Barber et al., 2023; Gibbs & Candès, 2022; Farinhas et al., 2023; Zou & Liu, 2024). When domain shifts occur, this assumption is violated and the coverage can drop substantially. Prior studies (Yilmaz & Heckel, 2022; Bhatnagar et al., 2023) show that even mild shifts cause sharp declines in coverage. This phenomenon is known as the **coverage gap** (Barber et al., 2023), defined as

$$\kappa = (1 - \alpha) - \mathbb{P}(y \in \mathcal{P}(x)), \tag{3}$$

where $1 - \alpha$ is the expected coverage and $\mathbb{P}(y \in \mathcal{P}(x))$ is the achieved coverage. Several extensions of CP attempt to compensate for this gap. NexCP (Barber et al., 2023) generalizes CP by employing weighted quantiles and a randomization technique, enabling robust predictive inference even when data exchangeability assumptions are violated. However, this method is designed for training phase and highly depends on a pre-defined domain shift value, which is not allowed in testing time. Moreover, QTC (Yilmaz & Heckel, 2022) recalibrate the quantile for coverage compensation. However, QTC suffers from the unreliable domain gap measurement in continual domain shifts and ignores the model differences. More details about existing non-exchangeable CP methods are discussed in Sec. 4.3. *This paper seeks to design a CP method for CTTA to act as an uncertainty indicator during testing time, and solve the coverage gap issue.*

## 4 CONFORMAL UNCERTAINTY INDICATOR FOR CTTA

### 4.1 CP WITH QUANTILE COMPENSATION

To close the coverage gap of conformal prediction under continual domain shifts, we propose the Conformal Uncertainty Indicator (CUI), a plug-and-play uncertainty estimator for CCTTA. CUI leverages a small labeled calibration set $\mathcal{C} = \{(x_1, y_1) \cdots, (x_{|\mathcal{C}|}, y_{|\mathcal{C}|})\}$ from the source domain and

adaptively adjusts the conformal quantile during testing. The key idea is to use both model and data differences to quantify domain shift, thereby correcting the prediction sets so that their size becomes a reliable measure of uncertainty. CUI is implemented in the following three steps.

**Step 1: Estimating joint domain shifts**

This step aims to obtain a reliable estimate of domain shifts under continual adaptation, which is essential for compensating the conformal quantile and mitigating the coverage gap. Existing extensions of CP have attempted to handle distribution shifts, but they remain inadequate for CTTA, as they often rely on assumptions that do not hold under CTTA (see Sec. 4.3 for more details). In particular, many approaches estimate domain discrepancy solely from the current model, which can be unreliable due to error accumulation. For example, prototype-based distances derived from the current model may no longer reflect the true data distribution once the model has drifted.

To obtain a more reliable estimate, we jointly consider *model shift* and *data shift*. Model shift reflects how much the current model $\theta^{\text{crt}}$ deviates from the source model $\theta^{\text{src}}$, while data shift captures how the test batch $\mathcal{B}$ diverges from the calibration set $\mathcal{C}$. We combine these perspectives by representing each sample $x$ with a joint probability vector that concatenates predictions from both models:

$$p(x) = \text{softmax}\left(\text{concat}(\pi_{\theta^{\text{src}}}(x), \pi_{\theta^{\text{crt}}}(x))\right). \tag{4}$$

The domain shift score $\rho$ is then obtained by comparing calibration and test samples via the Jensen–Shannon (JS) divergence:

$$\rho = \sum\nolimits_{x^{\text{calib}} \in \mathcal{C}} \sum\nolimits_{x^{\text{test}} \in \mathcal{B}} D_{\text{JS}}(p(x^{\text{test}})||p(x^{\text{calib}})). \tag{5}$$

We adopt JS divergence $D_{\text{JS}}$ as it is symmetric, bounded, and more stable than KL divergence, making it suitable for measuring discrepancies between probability distributions in dynamic test-time environments. This joint representation mitigates the bias from error accumulation and provides a more faithful measure of domain discrepancy. A larger $\rho$ indicates stronger distributional and model drift, and in the next step we show how this signal is used to compensate the conformal quantile to restore coverage.

**Step 2: Compensating the quantile threshold.**

The shift score $\rho$ obtained in Step 1 reflects how far the current test environment has drifted from the source distribution. Since larger shifts typically lead to a larger coverage gap, $\rho$ can be used as a proxy for the expected loss in coverage. In classical CP, the threshold conformal predictor (THR) (Sadinle et al., 2019) constructs prediction sets by thresholding non-conformity scores. Given a calibration set $\mathcal{C}$, the quantile threshold $\tau^*$ is defined as the $(1-\alpha)(\frac{|\mathcal{C}|+1}{|\mathcal{C}|})$-quantile of calibrated non-conformity scores:

$$\tau^* = \text{Quantile}(\mathcal{C}, (1-\alpha)) = \inf\left\{\tau : \mathbb{E}_{x \in \mathcal{C}}\mathbb{I}_{\{s(\pi(x))<\tau\}} \geq \frac{|\mathcal{C}|+1}{|\mathcal{C}|}(1-\alpha)\right\}. \tag{6}$$

For each calibration example, the non-conformity score is computed as

$$s(\pi_{\theta^{\text{crt}}}(x)) = 1 - \pi_{\theta^{\text{crt}}}(y|x), \tag{7}$$

that is, one minus the predicted probability of the true class. Intuitively, a smaller score corresponds to higher confidence in the correct label, while larger scores indicate greater uncertainty.

However, under continual domain shifts, $\tau^*$ becomes unreliable. Because the calibration distribution no longer matches the test distribution, $\tau^*$ is often too strict and leads to under-coverage. To mitigate this issue, we compensate the threshold using the shift score $\rho$:

$$\hat{\tau} = \tau^* + \beta \cdot \rho, \tag{8}$$

where $\beta$ is a tunable scaling factor. Increasing $\tau$ enlarges the prediction sets, thereby including more candidate labels and restoring coverage closer to the nominal $\alpha$ level. This compensation mechanism directly links the estimated shift to the degree of coverage correction, making the resulting prediction sets more trustworthy for uncertainty estimation in CTTA.

**Step 3: Computing the prediction set.**

Given the compensated threshold $\hat{\tau}$, we construct the prediction set for each test sample $x$ as

$$\mathcal{P}(x; \hat{\tau}) = \{y \in \mathcal{Y}^{\text{test}} \mid s(y|\pi(x)) < \hat{\tau}\}, \tag{9}$$

---

**Algorithm 1** Conformal Uncertainty Indicator in CTTA

---

**Input:** Test data point $x$, Pre-trained model $\pi$, calibration set $\mathcal{C}$, test data stream $\mathcal{X}^{\text{test}}$
1: Point prediction via the pre-trained model: $\hat{y} = \arg\max_{y \in \mathcal{Y}} \pi(y|x)$
2: Measure domain difference $\rho$ using Eq. (5)
3: Compute non-conformity scores for calibration set using Eq. (7)
4: Obtain the threshold $\tau^* = \text{Quantile}(\mathcal{C}, 1 - \alpha)$
5: Compensate threshold via $\hat{\tau} = \tau^* + \beta \cdot \rho$
6: Set prediction via threshold: $\mathcal{P}(x; \hat{\tau}) = \{y \in \mathcal{Y} | s(y|\pi(x)) < \hat{\tau}\}$
**Output:** Point prediction $\hat{y}$, Set prediction $\mathcal{P}$

---

where $s(y|\pi(x))$ denotes the non-conformity score of class $y$ under the current model prediction $\pi(x)$. The threshold $\hat{\tau}$ determines the size of the prediction set. A larger $\hat{\tau}$ allows more candidate labels to be included, resulting in larger sets, whereas a smaller $\hat{\tau}$ produces smaller sets. In this way the set size naturally represents prediction uncertainty. Large sets indicate that the model cannot confidently exclude many classes, while small sets correspond to more certain predictions. Compared with scalar confidence measures such as entropy, the set-based formulation of CP provides statistically valid coverage guarantees, which makes it more reliable in continual domain shift scenarios. The complete procedure of CUI is summarized in Algorithm 1.

### 4.2 CUI-GUIDED ADAPTATION

The size of the prediction set produced by CUI provides a natural indicator of reliability. A prediction set of size one implies that the model is confident about a single label, which we regard as the most reliable case. Larger sets indicate greater uncertainty, since the model cannot confidently rule out multiple alternatives. Empty prediction sets may occasionally occur under severe shifts, and we treat them as maximally unreliable. This allows CUI to guide adaptation and reduce the risk of error accumulation in CTTA.

We design a strategy that weights the contribution of each test sample according to its reliability. Samples with smaller prediction sets receive larger weights, which ensures that confident predictions play a stronger role in adaptation. Consider the case of Mean Teacher based adaptation (Wang et al., 2022; Brahma & Rai, 2023). The student is updated from the teacher's predictions, and the teacher is updated through exponential moving averaging (EMA) from the student. Under this setting, the CUI-guided student loss is defined as

$$L = -\mathbb{E}_{x \in \mathcal{B}} \gamma(x) \cdot \pi_{\boldsymbol{\theta}^{\text{tea}}}(x) \log \pi_{\boldsymbol{\theta}^{\text{stu}}}(x), \tag{10}$$

where $\boldsymbol{\theta}^{\text{tea}}$ and $\boldsymbol{\theta}^{\text{stu}}$ are the teacher and student models, respectively. The weight $\gamma(x)$ is determined by the relative size of the prediction set:

$$\gamma(x) = \begin{cases} \dfrac{\max_{x' \in \mathcal{B}}(|\mathcal{P}(x')|) - |\mathcal{P}(x)| + \delta}{\max_{x' \in \mathcal{B}}(|\mathcal{P}(x')|) - 1 + \delta}, & |\mathcal{P}(x)| > 0 \\ 0, & |\mathcal{P}(x)| = 0, \end{cases} \tag{11}$$

where $\delta$ is a small constant that avoids division by zero. This design normalizes weights within each mini-batch, so that reliable samples consistently dominate the update. When $|\mathcal{P}(x)| = 1$, we obtain $\gamma(x) = 1$, which corresponds to the most reliable case. Although we describe the approach using Mean Teacher, the weighting scheme is general and can be integrated into other CTTA frameworks. This makes CUI a flexible plug-in for reliability-aware adaptation.

### 4.3 DISCUSSION

**Comparison with existing non-exchangeable CP methods.** We compare our CUI with two recent non-exchangeable CP methods, including NexCP (Farinhas et al., 2023) and QTC (Yilmaz & Heckel, 2022). First, both NexCP and QTC are designed only for uncertainty indication instead of adaptation improvement. NexCP is designed for training time, where it specifies a constant to represent the domain difference from the source domain to the target domain. Specifically, NexCP directly compensates the coverage by

$$\mathbb{P}(y \in \mathcal{P}(x)) \geq 1 - \alpha - 2 \sum_{i=1}^{n} w_i \epsilon_i, \tag{12}$$

where $\epsilon_i$ is a predefined constant measure of how much the distribution has shifted from the test sample to the $i$-th calibrated sample and $w_i$ is a corresponding weight. NexCP will satisfy marginal coverage, and are exact when the magnitude of the distribution shift is known, which is infeasible in test time. In contrast, CUI is designed for testing, and measuring the distribution shifts adaptively.

QTC proposes to replace the user-specified $\alpha$ to a new coverage level $\beta_{\text{QTC}}$ calculated as

$$\beta_{\text{QTC}} = \min \left[ \mathbb{E}_{x \in \mathcal{C}} \mathbb{I}_{\{s(\pi(x)) < \text{Quantile}(\mathcal{B}, \alpha)\}}, 1 - \mathbb{E}_{x \in \mathcal{B}} \mathbb{I}_{\{s(\pi(x)) < \text{Quantile}(\mathcal{C}, 1-\alpha)\}} \right]. \quad (13)$$

Based on the current model $\pi$, QTC finds a threshold on the scores of the model on the unlabeled samples and predicts the coverage level by utilizing how the distribution of the scores changes across test distribution with respect to this threshold. However, QTC ignore the adaptation on continual domain shifts may suffer serious error accumulation, making the current model unreliable. This leads to the CP results being unreliable too. Instead, our CUI considers the error accumulation and evaluates domain shifts based on a joint distribution difference. More details are shown in Appendix.

**Calibration data in testing.** As defined in the CCTTA setting, a small labeled calibration buffer from the source domain is available at test time. This assumption is not unique to our work, since many related areas also rely on maintaining small data buffers. Many continual learning (Rolnick et al., 2019; Van de Ven et al., 2020) methods store and retrain previous training examples to avoid catastrophic forgetting of past tasks, named replay strategy. In comparison with replay, the calibration set in CUI is not used for adaptation but calibration in testing time, and the calibration set will not be updated in our method. Practical approaches in real-world settings involve storing samples to improve testing outcomes, such as Tomani et al. (2021) and Rahimi et al. (2020) leverage post-hoc calibration to achieve better performance under domain drift scenarios by using validation or calibration sets. In the CTTA tasks, some existing methods use source data to improve the adaptation such as Döbler et al. (2023). *The proposed CUI is plug-and-play, particularly well-suited for scenarios where the continuous accumulation of errors over long-term testing periods is unacceptable, such as in autonomous driving and medical applications.* In these contexts, proactively assessing model uncertainty is essential to ensure safety and reliability, and it is acceptable for users to maintain a small set of calibration data. Furthermore, for a fair comparison, *calibration sets are consistently employed across all methods discussed in the experiments.*

## 5 EXPERIMENT

### 5.1 EXPERIMENTAL SETTING

**Dataset**. We employ the CIFAR10-to-CIFAR10C, CIFAR100-to-CIFAR100C, and ImageNet-to-ImageNetC datasets as benchmarks to assess the effectiveness of CUI (CIFAR10C, CIFAR100C and ImageNetC for short). Each dataset comprises 15 distinct types of corruption, each applied at severity level of 5. These corruptions are applied to test images from the original datasets.

**Calibration Set Construction**. For each dataset, we construct a small labeled calibration buffer from the source domain in two possible ways: (i) splitting off a disjoint portion of source data before pretraining (***privacy-first***, which requires retraining), or (ii) reusing a small subset of the training data (***efficiency-first***, which avoids retraining). In our experiments, we adopt the efficiency-first strategy and set the calibration buffer sizes to 50, 100, and 500 for CIFAR10C, CIFAR100C, and ImageNetC, respectively. The buffer is fixed throughout testing and used solely for CP.

**Pretrained Model**. Following previous studies (Wang et al., 2020; 2022), we adopt pretrained WideResNet-28 (Zagoruyko & Komodakis, 2016) model for CIFAR10C, pretrained ResNeXt-29 (Xie et al., 2017) for CIFAR100C, and standard pretrained ResNet-50 (He et al., 2016) for ImagenetC. For a fair comparison, we conduct all experiments in a same environment.

**Evaluation Metric**: We use two kinds of metrics including testing performance, CP performance. We use $\hat{\mathcal{D}}$ to represent the testing data with labels. (1) For testing performance, we use the error rate (ERR) following existing CTTA methods (Wang et al., 2022). (2) For CP performance, we leverage coverage and inefficiency for joint evaluation:

$$\text{COV} = \mathbb{E}_{(x,y) \in \hat{\mathcal{D}}} \mathbb{I}(y \in \mathcal{P}(x)), \ \text{INE} = \mathbb{E}_{x \in \hat{\mathcal{D}}} |\mathcal{P}(x)|. \quad (14)$$

The coverage should be near to the user expectation and the inefficiency should be small but larger than 0. Specifically, COV closer to $1 - \alpha$ indicates a more effective uncertainty estimation of the CP. For example, with $\alpha = 0.1$, the COV should be close to $90\%$. INE, on the other hand, indicates

Table 1: Results of combining CUI with exiting CTTA methods on the three datasets. For calibration, the **privacy-first** strategy uses a disjoint split with a retrained source model, while the **efficiency-first** strategy reuses common-used pre-trained source model. For each SOTA method, the first line means the vanilla implementation only with CUI for uncertainty estimation, and the second line means the method uses uncertainty to guide the adaptation. *Because CUI does not change the ERR, we omit the results of these methods **w/o both CUI and CPAda** for saving space.*

| Method 1. CUI: Sec. 4.1  2. CPAda: Sec. 4.2 | Privacy First (Calibration data ∩ Training data = ∅) | | | | | | | | | Efficiency First (Calibration data ⊂ Training data) | | | | | | | | |
|---|---|---|---|---|---|---|---|---|---|---|---|---|---|---|---|---|---|---|
| | α = 0.3 | | | α = 0.2 | | | α = 0.1 | | | α = 0.3 | | | α = 0.2 | | | α = 0.1 | | |
| | ERR | COV | INE | ERR | COV | INE | ERR | COV | INE | ERR | COV | INE | ERR | COV | INE | ERR | COV | INE |
| **CIFAR10-CIFAR10C** | | | | | | | | | | | | | | | | | | |
| Tent + CUI | 21.65 | 69.12 | 0.89 | 21.65 | 78.45 | 1.96 | 21.65 | 87.93 | 2.33 | 20.45 | 68.55 | 0.81 | 20.45 | 77.88 | 1.02 | 20.45 | 87.67 | 1.57 |
| Tent + CUI + CPAda | 19.70 | 69.04 | 0.81 | 19.25 | 76.73 | 1.02 | 19.25 | 87.17 | 1.56 | 18.06 | 67.91 | 0.77 | 18.32 | 78.19 | 1.01 | 18.22 | 87.57 | 1.29 |
| CoTTA + CUI | 16.34 | 68.77 | 0.81 | 16.34 | 78.45 | 1.89 | 16.34 | 87.85 | 1.66 | 16.22 | 67.86 | 1.15 | 16.22 | 75.36 | 1.09 | 16.22 | 89.35 | 1.90 |
| CoTTA + CUI + CPAda | 15.73 | 68.77 | 0.81 | 15.75 | 77.93 | 1.03 | 15.71 | 87.02 | 1.46 | 15.52 | 66.62 | 0.81 | 15.73 | 77.25 | 1.00 | 15.65 | 88.53 | 1.61 |
| SATA + CUI | 16.31 | 68.25 | 0.75 | 16.31 | 77.78 | 0.95 | 16.31 | 86.07 | 1.24 | 16.13 | 68.28 | 0.84 | 16.13 | 77.14 | 0.85 | 16.13 | 85.61 | 1.09 |
| SATA + CUI + CPAda | 15.79 | 68.83 | 0.75 | 15.76 | 76.68 | 0.89 | 15.72 | 86.97 | 1.30 | 15.59 | 67.94 | 0.73 | 15.56 | 78.49 | 0.92 | 15.60 | 88.68 | 1.24 |
| RDumb + CUI | 18.31 | 68.62 | 0.76 | 18.31 | 78.82 | 0.94 | 18.31 | 85.60 | 1.15 | 17.63 | 68.37 | 0.76 | 17.63 | 77.87 | 0.91 | 17.63 | 86.23 | 1.17 |
| RDumb + CUI + CPAda | 16.73 | 73.55 | 0.83 | 16.73 | 79.30 | 0.94 | 16.81 | 86.41 | 1.18 | 16.23 | 68.30 | 0.74 | 16.31 | 76.63 | 0.87 | 16.33 | 84.38 | 1.09 |
| C-CoTTA + CUI | 14.99 | 68.39 | 0.73 | 14.99 | 78.42 | 1.23 | 14.99 | 86.92 | 1.75 | 14.74 | 66.16 | 0.70 | 14.74 | 77.46 | 0.87 | 14.74 | 87.52 | 1.44 |
| C-CoTTA + CUI + CPAda | 14.75 | 66.97 | 0.72 | 14.72 | 77.10 | 1.14 | 14.76 | 86.42 | 1.55 | 14.32 | 68.82 | 0.74 | 14.38 | 75.53 | 0.85 | 14.33 | 88.47 | 1.64 |
| RMT + CUI | 14.66 | 68.86 | 0.75 | 14.66 | 76.81 | 1.14 | 14.66 | 87.37 | 1.45 | 14.54 | 68.29 | 0.85 | 14.54 | 78.37 | 1.10 | 14.54 | 89.06 | 1.50 |
| RMT + CUI + CPAda | 14.33 | 66.53 | 0.72 | 14.36 | 78.04 | 1.22 | 14.44 | 86.29 | 1.26 | 14.28 | 69.17 | 0.83 | 14.31 | 77.28 | 0.91 | 14.25 | 86.58 | 1.70 |
| **CIFAR100-CIFAR100C** | | | | | | | | | | | | | | | | | | |
| Tent + CUI | 62.24 | 69.23 | 2.66 | 62.24 | 78.50 | 4.44 | 62.24 | 87.24 | 11.19 | 60.93 | 69.04 | 17.32 | 60.93 | 77.15 | 27.97 | 60.93 | 84.63 | 35.52 |
| Tent + CUI + CPAda | 46.50 | 68.88 | 1.53 | 46.56 | 76.85 | 3.68 | 45.95 | 87.42 | 4.13 | 49.87 | 68.22 | 20.66 | 52.90 | 78.93 | 24.34 | 51.56 | 84.48 | 28.61 |
| CoTTA + CUI | 36.41 | 68.08 | 1.86 | 36.41 | 77.01 | 2.82 | 36.41 | 87.31 | 4.96 | 32.50 | 66.59 | 2.42 | 32.50 | 78.39 | 5.11 | 32.50 | 88.68 | 11.58 |
| CoTTA + CUI + CPAda | 32.11 | 67.81 | 1.69 | 32.16 | 79.33 | 3.34 | 32.31 | 89.64 | 9.69 | 30.93 | 64.65 | 1.85 | 30.99 | 75.08 | 3.16 | 31.59 | 84.61 | 6.45 |
| SATA + CUI | 33.46 | 69.32 | 1.81 | 33.46 | 76.84 | 2.79 | 33.46 | 87.39 | 7.06 | 30.30 | 68.69 | 1.55 | 30.30 | 77.80 | 2.64 | 30.30 | 87.82 | 6.02 |
| SATA + CUI + CPAda | 32.38 | 68.36 | 1.65 | 32.39 | 77.85 | 2.92 | 32.46 | 89.51 | 8.64 | 29.14 | 68.81 | 1.44 | 28.94 | 76.29 | 2.08 | 28.78 | 84.92 | 3.69 |
| RDumb + CUI | 45.93 | 68.01 | 2.29 | 45.93 | 76.86 | 3.38 | 45.93 | 88.48 | 7.23 | 45.10 | 68.56 | 2.06 | 45.10 | 78.02 | 2.21 | 45.10 | 87.68 | 2.23 |
| RDumb + CUI + CPAda | 42.12 | 68.62 | 1.76 | 42.23 | 79.30 | 2.94 | 42.26 | 86.21 | 7.89 | 43.42 | 69.49 | 2.72 | 43.22 | 76.10 | 2.86 | 43.36 | 85.28 | 3.40 |
| C-CoTTA + CUI | 32.79 | 68.58 | 1.83 | 32.79 | 78.12 | 3.21 | 32.79 | 88.37 | 7.62 | 29.90 | 69.75 | 1.71 | 29.90 | 76.54 | 2.51 | 29.90 | 84.51 | 4.70 |
| C-CoTTA + CUI + CPAda | 31.52 | 68.08 | 1.66 | 31.44 | 77.96 | 2.97 | 31.47 | 88.19 | 7.20 | 29.31 | 68.79 | 2.46 | 29.28 | 78.64 | 2.60 | 29.17 | 86.08 | 5.32 |
| RMT + CUI | 32.53 | 68.37 | 1.45 | 32.53 | 77.06 | 2.75 | 32.53 | 88.48 | 7.46 | 29.00 | 69.41 | 1.69 | 29.00 | 76.71 | 2.62 | 29.00 | 87.97 | 5.80 |
| RMT + CUI + CPAda | 31.43 | 67.47 | 1.39 | 31.32 | 76.71 | 2.62 | 31.45 | 86.97 | 6.40 | 28.35 | 67.67 | 1.40 | 28.33 | 77.06 | 2.75 | 28.28 | 87.71 | 4.49 |
| **ImageNet-ImageNetC** | | | | | | | | | | | | | | | | | | |
| Tent + CUI | 63.69 | 68.12 | 43.09 | 63.69 | 78.07 | 114.50 | 64.69 | 87.42 | 265.59 | 62.60 | 69.09 | 47.80 | 62.60 | 79.40 | 82.62 | 62.60 | 88.48 | 163.09 |
| Tent + CUI + CPAda | 62.50 | 69.26 | 47.89 | 62.53 | 76.89 | 112.25 | 62.60 | 88.71 | 272.71 | 61.50 | 69.26 | 47.89 | 61.53 | 76.19 | 43.25 | 61.60 | 88.71 | 164.50 |
| CoTTA + CUI | 69.03 | 68.88 | 84.43 | 69.03 | 79.01 | 110.13 | 69.03 | 88.28 | 188.43 | 62.70 | 68.43 | 69.74 | 62.70 | 78.07 | 90.86 | 62.70 | 86.70 | 171.33 |
| CoTTA + CUI + CPAda | 67.56 | 67.74 | 80.13 | 67.42 | 78.42 | 114.43 | 67.32 | 89.04 | 179.64 | 61.22 | 69.01 | 69.32 | 61.30 | 77.42 | 86.23 | 61.24 | 87.40 | 172.24 |
| SATA + CUI | 61.81 | 69.83 | 81.31 | 61.81 | 76.97 | 118.13 | 61.81 | 87.95 | 212.59 | 60.10 | 69.38 | 75.93 | 60.10 | 77.42 | 120.44 | 60.10 | 88.12 | 218.29 |
| SATA + CUI + CPAda | 60.62 | 69.10 | 54.99 | 60.92 | 79.09 | 113.38 | 60.87 | 89.46 | 224.14 | 58.52 | 68.24 | 64.18 | 58.54 | 78.71 | 121.57 | 58.65 | 87.32 | 192.66 |
| RDumb + CUI | 64.46 | 66.68 | 21.67 | 64.46 | 79.49 | 57.39 | 64.46 | 88.55 | 156.87 | 62.45 | 67.54 | 23.38 | 62.45 | 79.22 | 67.03 | 62.45 | 88.83 | 147.44 |
| RDumb + CUI + CPAda | 62.25 | 67.29 | 22.74 | 62.29 | 78.28 | 56.45 | 62.18 | 87.34 | 152.11 | 60.26 | 67.57 | 24.52 | 60.32 | 78.39 | 62.16 | 60.54 | 89.01 | 156.74 |
| C-CoTTA + CUI | 60.42 | 68.11 | 36.13 | 60.42 | 75.19 | 32.61 | 60.42 | 87.70 | 91.22 | 59.40 | 67.45 | 17.09 | 59.40 | 78.14 | 39.26 | 59.40 | 88.09 | 100.20 |
| C-CoTTA + CUI + CPAda | 59.48 | 68.03 | 20.87 | 59.52 | 77.24 | 42.90 | 59.53 | 88.74 | 96.05 | 58.36 | 68.31 | 18.73 | 58.33 | 79.05 | 40.46 | 58.39 | 87.67 | 98.40 |
| RMT + CUI | 61.64 | 69.79 | 19.44 | 61.64 | 78.05 | 37.59 | 61.64 | 86.15 | 82.13 | 59.80 | 69.53 | 18.73 | 59.80 | 78.04 | 38.18 | 59.80 | 86.83 | 82.37 |
| RMT + CUI + CPAda | 59.62 | 69.71 | 18.98 | 59.65 | 78.57 | 39.07 | 59.66 | 86.04 | 76.99 | 59.28 | 69.57 | 19.30 | 59.25 | 76.91 | 34.27 | 59.30 | 87.35 | 87.60 |

lower uncertainty when closer to 1, while values closer to 0 suggest that no valid prediction. INE greater than 1, with larger values indicating higher uncertainty.

## 5.2 MAJOR RESULTS

CUI is a play-and-plug uncertainty indicator. To evaluate the effect of CUI, we select several well-known and state-of-the-art methods as the baseline methods, including TENT (Wang et al., 2020), CoTTA (Wang et al., 2022), SATA (Chakrabarty et al., 2023), RMT (Döbler et al., 2023), C-CoTTA (Shi et al., 2024) and RDumb (Press et al., 2024). All compared methods adopt the same pre-trained model under the same calibration set construction strategy. For each selected method, we use the proposed CUI for uncertainty measurement, and based on this, we compare two results: one without adaptation and one using CUI guidance for domain adaptation. These two results are represented as adjacent rows in the table, such as "CoTTA+CUI" and "CoTTA+CUI+CPAda". We use three expected coverage factors $\alpha = 0.1, 0.2, 0.3$, which represent that the user would like $90\%, 80\%, 70\%$ coverage for the prediction. The results are shown in Table 1. First, with the inclusion of CUI, it is possible to estimate uncertainty (INE) that closely aligns with the predefined $\alpha$ values. In most cases, when CPAda is not employed, the INE values reveal significant inherent uncertainties within the baseline method. These uncertainties are associated with the dataset that more complex datasets typically exhibit higher INE values. Moreover, the INE varies depending on the $\alpha$ value. Specifically, smaller $\alpha$ values correspond to larger INE, as smaller $\alpha$ thresholds demand higher fault tolerance. This relationship highlights the trade-off between the level of certainty required and the algorithm's ability to meet that requirement. Second, the integration of CUI-guided CPAda improves existing methods, reducing ERR and lowering INE, indicating more accurate and confident predictions. Finally, the comparison between Privacy-First and Efficiency-First strategies shows minimal performance differences, suggesting that users can select the calibration dataset construction method based on their specific application needs without compromising results.

Table 2: Comparisons with non-exchangeable CP methods.

| | | Privacy First | | | | | | Efficiency First | | | | | |
| | | w/o CPAda | | | w/ CPAda | | | w/o CPAda | | | w/ CPAda | | |
| $\alpha$ | CP Method | ERR | COV | INE | ERR | COV | INE | ERR | COV | INE | ERR | COV | INE |
|---|---|---|---|---|---|---|---|---|---|---|---|---|---|
| | N/A | 35.15 | 23.39 | 0.28 | - | - | - | 32.77 | 34.27 | 0.44 | - | - | - |
| 0.3 | THR Sadinle et al. (2019) | 35.15 | 23.39 | 0.28 | 35.18 | 21.31 | 0.24 | 32.72 | 34.17 | 0.44 | 31.89 | 39.81 | 0.50 |
| | NexCP Farinhas et al. (2023) | 35.15 | 23.46 | 0.28 | 35.21 | 21.75 | 0.25 | 32.72 | 34.68 | 0.45 | 31.70 | 40.31 | 0.51 |
| | QTC Yilmaz & Heckel (2022) | 35.15 | 40.70 | 0.59 | 33.79 | 42.25 | 0.59 | 32.72 | 52.15 | 0.87 | 31.00 | 53.13 | 0.75 |
| | SaoCP Bhatnagar et al. (2023) | 35.15 | 35.87 | 0.49 | 34.16 | 41.41 | 0.58 | 32.72 | 62.97 | 1.43 | 29.71 | 68.20 | 1.30 |
| | CUI | 35.15 | 69.64 | 2.70 | **32.76** | 68.02 | 2.18 | 32.72 | 68.95 | 2.01 | **29.48** | 68.07 | 2.17 |
| 0.2 | THR Sadinle et al. (2019) | 35.15 | 29.05 | 0.37 | 34.80 | 27.93 | 0.34 | 32.72 | 42.17 | 0.60 | 31.43 | 48.32 | 0.67 |
| | NexCP Farinhas et al. (2023) | 35.15 | 29.42 | 0.37 | 34.78 | 28.39 | 0.35 | 32.72 | 41.87 | 0.59 | 31.32 | 48.36 | 0.66 |
| | QTC Yilmaz & Heckel (2022) | 35.15 | 46.63 | 0.75 | 33.54 | 47.84 | 0.73 | 32.72 | 59.96 | 1.22 | 30.53 | 61.53 | 0.99 |
| | SaoCP Bhatnagar et al. (2023) | 35.15 | 44.01 | 0.68 | 33.71 | 51.35 | 0.86 | 32.72 | 71.15 | 2.24 | 29.36 | 74.98 | 1.78 |
| | CUI | 35.15 | 77.58 | 4.60 | **32.59** | 77.46 | 3.64 | 32.72 | 76.73 | 3.42 | **29.17** | 79.15 | 2.27 |
| 0.1 | THR Sadinle et al. (2019) | 35.15 | 37.25 | 0.52 | 34.20 | 37.12 | 0.49 | 32.72 | 53.69 | 0.95 | 30.64 | 59.89 | 0.97 |
| | NexCP Farinhas et al. (2023) | 35.15 | 37.71 | 0.53 | 34.17 | 37.70 | 0.51 | 32.72 | 53.17 | 0.92 | 30.62 | 59.83 | 0.97 |
| | QTC Yilmaz & Heckel (2022) | 35.15 | 55.56 | 1.10 | 33.25 | 54.29 | 0.93 | 32.72 | 69.14 | 1.92 | 29.58 | 72.31 | 1.50 |
| | SaoCP Bhatnagar et al. (2023) | 35.15 | 53.28 | 0.99 | 33.30 | 59.00 | 1.16 | 32.72 | 83.51 | 5.64 | 29.37 | 80.99 | 2.47 |
| | CUI | 35.15 | 86.41 | 9.30 | **32.74** | 89.02 | 11.48 | 32.72 | 86.38 | 7.78 | **29.17** | 88.35 | 5.47 |

## 5.3 MORE ANALYSIS ON THE PROPOSED METHOD

**Comparisons with non-exchangeable CP methods**. In Table 2, we compare our CUI with other CP methods including THR Sadinle et al. (2019), NexCP Barber et al. (2023) and QTC Yilmaz & Heckel (2022). THR is an exchangeable CP method and never considers domain shifts in CTTA, thus it obtains an obvious coverage gap. NexCP and QTC are two non-exchangeable methods, with detailed comparisons available in Sec. 4.3. First, for NexCP, we use the same fixed value for domain shift estimation as in the original paper, and NexCP is only slightly better than THR and struggles to estimate domain differences in advance during testing. Then, although QTC estimates domain differences in real time, it neglects the unreliability of the current model due to error accumulation over long testing periods. This method yields better results than both THR and NexCP. *However, these methods all suffer from coverage gap issues, and the uncertainty estimation is unreliable in CTTA, even if their INE is close to 1.* Instead, CUI obtains near-expected coverage when estimating testing uncertainty. Next, we compare our domain adaptation method (CPAda) using different CP techniques that are similar to the proposed method, and the results show that CUI provides better guidance for adaptation and obtains lower error rates.

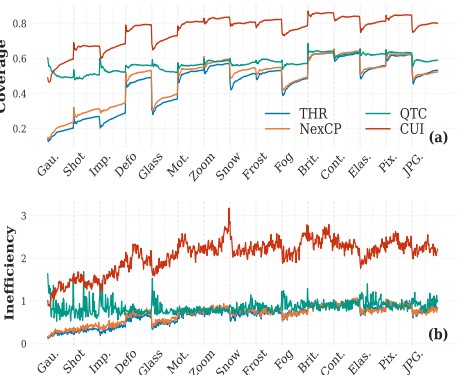

Figure 2: Changes of COV and INE.

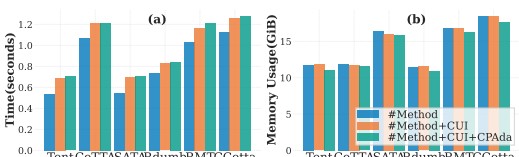

Figure 3: Hyperparameters on CIFAR100C.

Figure 4: Time and memory cost on CIFAR100C.

**Coverage and Inefficiency changes in CTTA**. In Fig. 2, we show the coverage and inefficiency changes of different CP methods. As shown in Fig. 2(a), coverage varies significantly across methods, reflecting domain disparities. Existing methods, such as THR and NexCP, show notable coverage gaps, while QTC performs well initially but struggles with error accumulation. In contrast, CUI achieves comparable initial coverage to QTC and surpasses it in later domains. Fig. 2(b) illustrates inefficiency trends, revealing that existing methods, despite low coverage, fail to account for error accumulation during domain shifts, leading to overconfidence. CUI, however, captures this accumulation, with inefficiency increasing as domains change, reflecting growing uncertainty. When CUI guides domain adaptation, inefficiency decreases, demonstrating effective uncertainty control.

**Storage analysis and comparison with replay strategy**. As discussed in Sec. 4.3, CP-based methods need to maintain an extra calibration set for uncertainty estimation. Although effectively measuring uncertainty is crucial in testing systems, using CP requires a certain amount of memory

Table 3: Comparison with replay strategy.

| Method | Storage | Privacy First | | | Efficiency First | | |
|--------|---------|------|------|------|------|------|------|
| | | ERR | COV | INE | ERR | COV | INE |
| Baseline | | 35.15 | 23.39 | 0.28 | 32.77 | 34.27 | 0.44 |
| Soure Replay CUI + CPAda | 100 | 35.03 | 21.88 | 0.25 | 32.64 | 7.47 | 0.08 |
| | | 32.59 | 78.11 | 3.29 | 29.17 | 79.15 | 2.27 |
| Soure Replay CUI + CPAda | 200 | 35.02 | 13.34 | 0.14 | 32.52 | 8.20 | 0.09 |
| | | 31.97 | 79.02 | 7.61 | 29.38 | 77.05 | 2.04 |
| Soure Replay CUI + CPAda | 300 | 34.22 | 13.74 | 0.15 | 32.09 | 8.97 | 0.09 |
| | | 31.33 | 78.59 | 5.12 | 29.77 | 77.48 | 2.18 |

Table 4: Comparison on DomainNet-126.

| Method | real | clipart | painting | sketch | Mean |
|--------|------|---------|----------|--------|------|
| Tent | 43.88 | 45.48 | 38.23 | 38.54 | 41.53 |
| Tent+CUI + CPAda | 41.70 | 44.78 | 36.90 | 36.72 | 40.03 |
| CoTTA | 43.00 | 42.80 | 36.85 | 37.04 | 39.92 |
| CoTTA+CUI + CPAda | 41.32 | 40.17 | 35.42 | 34.81 | 37.93 |
| RMT | 38.86 | 39.22 | 33.31 | 33.63 | 36.26 |
| RMT+CUI + CPAda | 37.34 | 37.82 | 31.23 | 32.06 | 34.61 |

storage. We analyze the impact of this storage on performance in Table 3 and find that a larger storage capacity leads to better CP performance, as more calibration data provides a more accurate representation of the original data distribution. Additionally, we compare CUI with a classic storage method in continual learning, the source replay strategy, where we use the same samples for replay when conducting adaptation. We find that CUI achieves better accuracy while maintaining the same amount of stored data, which shows the significance of reducing error accumulation in CTTA.

**Impacts of user-specified coverage level $\alpha$.** In CP, we have a user-specified coverage level $\alpha \in (0, 1)$ (Eq. (2)), which is generally considered to represent a user pre-specified error rate. In Fig. 3(a), we show that the infuence of different $\alpha$ from $0.1$ to $0.9$. The results show that a large $\alpha$ means that the user accepts a lower coverage rate, reflecting a large error rate.

**Analysis of compensation factor $\beta$.** We also analyze the influence of different compensation factors $\beta$ in Eq. (8), which represents the compensation level. The results are shown in Fig. 3(b), we find that small $\beta$ decrease the compensation performance and large $\beta$ may result in overcompensation.

**Time and memory cost**. We analyze CUI's impact on time and memory cost increases compared to the original methods, as shown in Fig. 4. It is evident that our CUI and CPAda strategies slightly increase implementation time due to the forward propagation of calibration data. However, CPAda reduces memory costs by performing backpropagation only on selected samples.

**Error comparison on DomainNet (Peng et al., 2019).** DomainNet is a commonly used domain shift dataset in the traditional TTA task. We evaluate the proposed method on DomainNet. As shown in Table 4, our method can improve Tent, CoTTA, and RMT by introducing uncertainty estimation.

Table 5: Comparison on Small Batch Sizes.

| Method | 100 | 50 | 10 |
|--------|-----|-----|-----|
| Tent | 22.06 | 28.66 | 75.34 |
| Tent+CUI + CPAda | 19.75 | 22.38 | 72.09 |
| CoTTA | 18.27 | 20.43 | 57.25 |
| CoTTA+CUI + CPAda | 16.52 | 18.33 | 52.75 |
| SATA | 16.45 | 16.90 | 20.72 |
| SATA+CUI + CPAda | 15.77 | 16.23 | 20.34 |

Table 6: Comparison on Different corruption orders.

| Method | 1 | 2 | 3 | 4 | 5 | avg | std |
|--------|-----|-----|-----|-----|-----|------|------|
| Tent | 20.45 | 20.08 | 18.73 | 19.27 | 21.65 | 20.04 | 1.13 |
| Tent+CUI + CPAda | 18.06 | 19.22 | 18.01 | 18.65 | 18.35 | 18.46 | 0.50 |
| CoTTA | 16.22 | 16.33 | 16.80 | 16.53 | 16.68 | 16.51 | 0.24 |
| CoTTA+CUI + CPAda | 15.52 | 15.23 | 15.46 | 15.51 | 15.58 | 15.46 | 0.14 |
| SATA | 16.13 | 16.18 | 16.42 | 16.16 | 16.27 | 16.23 | 0.12 |
| SATA+CUI + CPAda | 15.59 | 15.49 | 15.72 | 15.53 | 15.79 | 15.62 | 0.13 |

**Sensitivity to batch size.** We further evaluate different batch sizes $\{100, 50, 10\}$ in Table 5. Performance decreases for all methods as the batch size becomes smaller, yet CUI consistently improves robustness across settings, showing no additional instability under small-batch adaptation.

**Sensitivity to corruption order.** To assess robustness against different corruption sequences, we evaluated 5 randomly sampled orders (Table 6). Compared with other methods, CUI consistently reduced error rate, demonstrating improved stability under varying corruption orders.

# 6 CONCLUSION

We studied uncertainty estimation for CTTA under a calibration-aided setting. We proposed the CUI, which leverages a small labeled calibration buffer with conformal prediction. CUI jointly measures model shift and data shift to correct conformal quantiles, closes the coverage gap under domain shifts, and yields prediction sets whose size serves as a calibrated indicator of test-time uncertainty. We further introduced a CUI-guided adaptation strategy that updates models only on confident samples and improves the robustness of existing CTTA baselines. Experiments on three benchmarks validate that CUI provides reliable uncertainty estimates and enhances downstream adaptation. CUI requires a calibration buffer from the source domain, which may not always be available, and it currently operates at the instance level, limiting direct application to fine-grained tasks such as pixel-level segmentation. Future work will explore relaxing the reliance on calibration data through online or privacy-preserving calibration and extending CUI to structured outputs and dense prediction tasks.

ETHICS STATEMENT

This work does not involve human subjects, sensitive data, or other aspects that may raise ethical concerns. Therefore, we believe there are no ethical issues associated with our study.

REPRODUCIBILITY STATEMENT

We provide the source code in the supplementary zip to ensure reproducibility. The paper includes detailed descriptions of dataset construction, algorithmic designs, and experimental procedures, allowing others to reproduce and verify our results.

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

## A  COVERAGE PROOF IN CONFORMAL PREDICTION

### A.1  COVERAGE IN EXCHANGEABLE CONFORMAL PREDICTION (WITHOUT DOMAIN SHIFT)

**Theorem 1** (Exchangeable Conformal Prediction (Vovk et al., 2005))**.** *Assume the calibration set $\mathcal{C}$ and a new data sample $x$ are i.i.d. (or more generally, exchangeable), and the model $\pi$ treats the input data points symmetrically. Given a specified coverage level $\alpha$, the quantile can be calculated by*

$$\tau^* = \text{Quantile}[\mathcal{C}, (1-\alpha)] = \inf\left\{\tau \colon \frac{1}{|\mathcal{C}|}\sum_{x\in\mathcal{C}}\mathbb{I}_{\{s(\pi(x))<\tau\}} \geq \frac{|\mathcal{C}|+1}{|\mathcal{C}|}(1-\alpha)\right\}. \quad (15)$$

*Then, the conformal prediction set is defined as*

$$\mathcal{P}(x) = \{y|s(\pi(x)) < \tau^*\}, \quad (16)$$

*and satisfies*

$$\mathbb{P}(y \in \mathcal{P}(x)) \geq 1 - \alpha. \quad (17)$$

**Proof**. The coverage proof of exchangeable CP is following Barber et al. (2023). First, we define the strange data points in the calibration set as an index set:

$$\mathcal{S} = \{i \in [1, n+1] : s(\pi(x_i)) > \tau^*\} \quad (18)$$

The strange points are with the largest $\lfloor \alpha(n+1) \rfloor$ non-conformity score. Because of the definition of quantile, it is easy to find that

$$|\mathcal{S}| \leq \alpha(n+1). \quad (19)$$

Then, for a test sample $x_{n+1}$, if it was failed-coverage, say $\hat{y}_{n+1} \notin \mathcal{P}(x_{n+1})$, this means that $s(\pi(x_i)) > \tau^*$. Thus, we have the strange probability:

$$\begin{aligned} p(y_{n+1} \notin \mathcal{P}(x_{n+1})) &= p(n+1 \in \mathcal{S}) \\ &= \mathbb{E}_{i\in[1,n+1]}p(i \in \mathcal{S}) \\ &= \frac{|\mathcal{S}|}{n+1} \end{aligned} \quad (20)$$

Because of the exchangeability assumation, we have

$$p(y_{n+1} \notin \mathcal{P}(x_{n+1})) \leq \alpha \quad (21)$$

The coverage of exchangeable conformal prediction is obtained proof. □

### A.2  COVERAGE IN NON-EXCHANGEABLE CONFORMAL PREDICTION (WITH DOMAIN SHIFTS)

In this subsection, we prove that why the proposed method can be used to compensate coverage gap in CP when domain shifts. First, following Barber et al. (2023), we give the lower bound of the coverage in non-exchangeable CP when the domain shifts is known.

**Lemma 1** (Coverage gap upper bound). *Assume that $\forall x \in \mathcal{C}$ and $x^{\text{test}}$ are independent. In a CP approach, the coverage gap can be bounded by the following inequality:*

$$\kappa = (1 - \alpha) - \mathbb{P}\{y \in \mathcal{P}(x)\}$$

$$\leq \frac{2}{n+1} \sum_{i=1}^{n} w_i \cdot \mathrm{d}_{\text{TV}}\left[(x_i, y_i), (x^{\text{test}}, y^{\text{test}})\right], \tag{22}$$

*where $\mathrm{d}_{\text{TV}}$ is a total variation distance. $w_i$ is a prespecified importance weight for the $i$-th calibration sample, and is set to $1$ in general CP.*

**Proof.** Let $\mathcal{X} = \mathcal{C} \cup \{(x^{\text{test}}, y^{\text{test}})\}$. Because $\forall x \in \mathcal{C}$ and $x^{\text{test}}$ are independent, we have

$$\kappa = (1 - \alpha) - \mathbb{P}\{y \in \mathcal{P}(x)\}$$

$$\leq \frac{1}{n+1} \sum_{i=1}^{n+1} w_i \cdot \mathrm{d}_{\text{TV}}\left(\mathcal{X}, (x_1, y_i)\right)$$

$$\leq \frac{1}{n+1} \sum_{i=1}^{n} w_i \cdot \left(2\mathrm{d}_{\text{TV}}\left[(x_i, y_i), (x^{\text{test}}, y^{\text{test}})\right]\right. \tag{23}$$

$$\left. - \mathrm{d}_{\text{TV}}\left[(x_i, y_i), (x^{\text{test}}, y^{\text{test}})\right]^2\right)$$

$$\leq \frac{2}{n+1} \sum_{i=1}^{n} w_i \cdot \mathrm{d}_{\text{TV}}\left[(x_i, y_i), (x^{\text{test}}, y^{\text{test}})\right],$$

where the second inequality can be obtained by the maximal coupling theorem (Den Hollander, 2012). That is, for two independent random variables $x$ and $y$, if we have another two independent random variables $\hat{x}$ and $\hat{y}$ and $(\hat{x}, \hat{y})$ is a maximal coupling for $(x, y)$, then we have $d_{\text{TV}}(x, y) = p(\hat{x} \neq \hat{y})$. $\qquad\square$

**Theorem 2** (Exchangeable Conformal Prediction with Known Shifts (Barber et al., 2023)). *Assume the calibration set $\mathcal{C}$ is i.i.d., but a new data sample $x$ is drawn from a different distribution. Given a specified coverage level $\alpha$, the quantile can be calculated by*

$$\tau^* = \text{Quantile}[\mathcal{C}, (1 - \alpha)] = \inf\left\{\tau : \frac{1}{|\mathcal{C}|} \sum_{x \in \mathcal{C}} \mathbb{I}_{\{s(\pi(x)) < \tau\}} \geq \frac{|\mathcal{C}| + 1}{|\mathcal{C}|}(1 - \alpha)\right\}. \tag{24}$$

*Then, the conformal prediction set is defined as*

$$\mathcal{P}(x) = \{y | s(\pi(x)) < \tau^*\}, \tag{25}$$

*and satisfies a coverage lower bound:*

$$\mathbb{P}(y \in \mathcal{P}(x)) \geq 1 - \alpha - \frac{2}{n} \sum_{i=1}^{n} w_i \cdot \mathrm{d}_{\text{TV}}\left[(x_i, y_i), (x^{\text{test}}, y^{\text{test}})\right]. \tag{26}$$

**Proof.** This theorem can be easily obtained from Lemma 1.

### A.3 COVERAGE OF CUI WITH DOMAIN SHIFTS

However, Theorem 2 is only appropriate for known domain difference. When the domain differences are unknown in test time, it is difficult to obtain a certain coverage lower bound. This explains why NexCP performs poorly in the CTTA task. QTC has designed a dynamic method for estimating domain differences, making it more suitable for testing compared to NexCP. However, the CTTA task requires multiple domain changes, which significantly impacts the model's ability to estimate domain differences due to error accumulation. Specifically, we compute the joint distribution difference of current data and calibration data between the source and current models.

In CUI, we dynamically evaluate the domain difference between the source data and the current test data. To mitigate the effect of error accumulation, we consider both model and data difference. We use the Jensen-Shannon (JS) divergence as the metric. Joint feature representation captures

correlations between different features, providing a more holistic view of the data distribution and how different models process it. The joint distribution can better reflect subtle differences between domains, enhancing the precision of JS divergence measures. Moreover, comparing joint feature distributions allows for a more detailed assessment of how much the current model has gained compared to the source model.

## B  UNCERTAINTY EVALUATION USING OTHER METRICS

In the main paper, we use two kinds of metrics including testing performance, CP performance. We use $\hat{\mathcal{D}}$ to represent the testing data with labels. (1) For testing performance, we use the error rate (ERR) following existing CTTA methods Wang et al. (2022) and the small, the better. (2) For CP performance, we leverage coverage and inefficiency for joint evaluation. The coverage should be near to the user expectation and the inefficiency should be small but larger than 0.

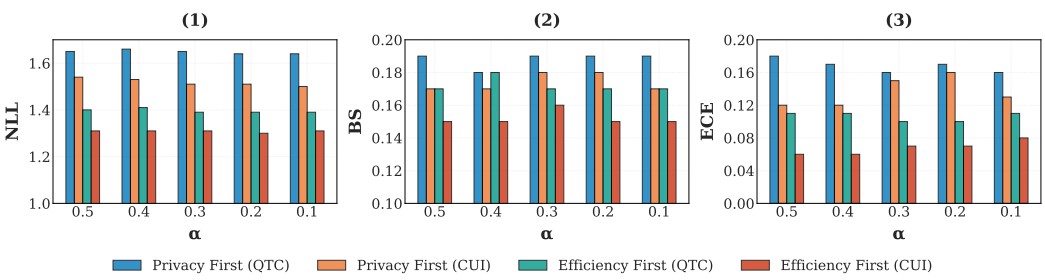

Figure 5: Uncertainty comparisons with QTC on CIFAR100-to-CIFAR100C using NLL,BS and ECE metrics.

In this subsection, for uncertainty measure, we use Negative Log Likelihood (NLL), Brier Score (BS, Brier (1950)) and Expected Calibration Error (ECE, Naeini et al. (2015)):

$$
\begin{aligned}
\text{NLL} &= -\mathbb{E}_{(x,y)\in\hat{\mathcal{D}}}\log(p(y|x)), \\
\text{BS} &= \mathbb{E}_{(x,y)\in\hat{\mathcal{D}}}\left(p(x) - 1(y)\right)^2, \\
\text{ECE} &= \sum_{i=1}^{10}\frac{|\mathcal{B}_i|}{|\hat{\mathcal{D}}|}\left|\text{acc}(\mathcal{B}_i) - \text{conf}(\mathcal{B}_i)\right|,
\end{aligned}
\tag{27}
$$

where $1(\cdot)$ means onehot. In ECE, we split samples into 10 bins by probability, and $\text{acc}(\mathcal{B}_i)$ means the bin accuracy and $\text{conf}(\mathcal{B}_i)$ is the mean confidence of the bin. The three metrics NLL, BS, and ECE are always used in scenarios where the true labels are known, which is impossible at test time and thus cannot be used as an uncertainty indicator. We compare CUI with QTC in Fig. 5, and find our method still outperforms this non-exchangeable CP method in these traditional uncertainty measures.

## C  DETAILED RESULTS

In our experiments, we employ the CIFAR10C, CIFAR100C, and ImageNetC datasets as benchmarks to assess the robustness of classification models. Each dataset comprises 15 distinct types of corruption, each applied at five different levels of severity (from 1 to 5). These corruptions are systematically applied to test images from the original CIFAR10 and CIFAR100 datasets, as well as validation images from the original ImageNet dataset. The 15 types of corruption are Gaussian, Shot, Impulse, Defocus, Glass, Motion, Zoom, Snow, Frost, Fog, Brightness, Contrast, Elastic, Pixelate, Jpeg. We show the detailed error results for each type of corruption in Tables 7, 8 and 9.

Table 7: Classification error rate (%) for the standard CIFAR10-to-CIFAR10C CTTA task. All results are evaluated with the largest corruption severity level 5 in an online fashion.

| Strategy | α | Method | Gau. | Sho. | Imp. | Def. | Gla. | Mot. | Zoo. | Sno. | Fro. | Fog | Bri. | Con. | Ela. | Pix. | Jpe. | Avg. |
|---|---|---|---|---|---|---|---|---|---|---|---|---|---|---|---|---|---|---|
| **Privacy First** | 0.3 | Tent+CUI | 24.73 | 20.82 | 29.39 | 14.71 | 33.20 | 17.38 | 14.78 | 19.77 | 19.38 | 19.67 | 14.66 | 19.11 | 26.33 | 22.13 | 28.62 | 21.65 |
| | | Tent+CUI+CPAda | 24.57 | 20.23 | 28.01 | 14.08 | 31.51 | 16.50 | 14.40 | 17.77 | 16.76 | 17.60 | 11.65 | 15.10 | 23.75 | 19.59 | 23.91 | 19.70 |
| | | CoTTA+CUI | 24.42 | 21.82 | 26.07 | 11.78 | 27.19 | 12.63 | 10.69 | 15.09 | 14.29 | 12.44 | 7.65 | 10.88 | 18.32 | 13.88 | 17.97 | 16.34 |
| | | CoTTA+CUI+CPAda | 23.09 | 20.63 | 25.84 | 10.51 | 26.39 | 11.53 | 10.26 | 14.05 | 14.08 | 12.52 | 7.72 | 10.44 | 17.18 | 13.71 | 18.00 | 15.73 |
| | | SATA+CUI | 23.31 | 19.63 | 28.16 | 11.59 | 28.10 | 12.55 | 10.42 | 14.03 | 13.69 | 12.27 | 7.64 | 10.69 | 18.92 | 14.16 | 19.53 | 16.31 |
| | | SATA+CUI+CPAda | 22.92 | 18.39 | 26.45 | 11.42 | 27.36 | 12.38 | 9.96 | 13.81 | 13.20 | 12.05 | 7.53 | 10.38 | 18.91 | 13.37 | 18.79 | 15.79 |
| | | RDumb+CUI | 24.06 | 19.81 | 27.56 | 12.93 | 29.55 | 15.10 | 12.78 | 16.67 | 16.57 | 15.52 | 9.48 | 13.27 | 21.67 | 17.48 | 22.25 | 18.31 |
| | | RDumb+CUI+CPAda | 23.27 | 20.53 | 25.21 | 11.47 | 27.44 | 12.81 | 11.50 | 15.58 | 14.92 | 13.24 | 8.31 | 10.75 | 20.51 | 15.55 | 19.89 | 16.73 |
| | | C-CoTTA+CUI | 23.43 | 17.75 | 23.26 | 11.85 | 24.35 | 12.61 | 10.35 | 13.73 | 12.53 | 11.95 | 8.37 | 9.82 | 16.46 | 12.10 | 16.32 | 14.99 |
| | | C-CoTTA+CUI+CPAda | 21.82 | 17.30 | 23.61 | 11.81 | 24.48 | 12.51 | 10.21 | 13.02 | 12.32 | 11.93 | 7.96 | 9.68 | 16.57 | 12.32 | 15.76 | 14.75 |
| | | RMT+CUI | 22.62 | 18.89 | 25.36 | 10.27 | 24.94 | 11.58 | 10.14 | 13.37 | 12.67 | 11.49 | 8.19 | 9.87 | 15.28 | 11.26 | 14.53 | 14.66 |
| | | RMT+CUI+CPAda | 22.09 | 17.86 | 23.93 | 10.63 | 23.82 | 11.83 | 10.49 | 12.79 | 12.51 | 11.07 | 8.33 | 9.56 | 15.10 | 11.04 | 13.94 | 14.33 |
| | 0.2 | Tent+CUI | 24.73 | 20.82 | 29.39 | 14.71 | 33.20 | 17.38 | 14.78 | 19.77 | 19.38 | 19.67 | 14.66 | 19.11 | 26.33 | 22.13 | 28.62 | 21.65 |
| | | Tent+CUI+CPAda | 24.77 | 20.67 | 28.34 | 13.73 | 30.32 | 16.01 | 13.88 | 17.65 | 16.73 | 16.03 | 10.48 | 13.61 | 23.16 | 19.26 | 24.10 | 19.25 |
| | | CoTTA+CUI | 24.42 | 21.82 | 26.07 | 11.78 | 27.19 | 12.63 | 10.69 | 15.09 | 14.29 | 12.44 | 7.65 | 10.88 | 18.32 | 13.88 | 17.97 | 16.34 |
| | | CoTTA+CUI+CPAda | 23.05 | 20.33 | 25.44 | 10.61 | 26.29 | 11.53 | 10.46 | 15.05 | 14.08 | 12.22 | 7.72 | 10.44 | 17.18 | 13.91 | 18.00 | 15.75 |
| | | SATA+CUI | 23.31 | 19.63 | 28.16 | 11.59 | 28.10 | 12.55 | 10.42 | 14.03 | 13.69 | 12.27 | 7.64 | 10.69 | 18.92 | 14.16 | 19.53 | 16.31 |
| | | SATA+CUI+CPAda | 22.92 | 18.27 | 26.45 | 11.43 | 27.23 | 12.17 | 9.96 | 13.76 | 13.20 | 12.15 | 7.53 | 10.28 | 18.91 | 13.36 | 18.79 | 15.76 |
| | | RDumb+CUI | 24.06 | 19.81 | 27.56 | 12.93 | 29.55 | 15.10 | 12.78 | 16.67 | 16.57 | 15.52 | 9.48 | 13.27 | 21.67 | 17.48 | 22.25 | 18.31 |
| | | RDumb+CUI+CPAda | 23.27 | 20.53 | 25.21 | 11.47 | 27.44 | 12.81 | 11.50 | 15.58 | 14.92 | 13.24 | 8.31 | 10.75 | 20.51 | 15.55 | 19.89 | 16.73 |
| | | C-CoTTA+CUI | 23.43 | 17.75 | 23.26 | 11.85 | 24.35 | 12.61 | 10.35 | 13.73 | 12.53 | 11.95 | 8.37 | 9.82 | 16.46 | 12.10 | 16.32 | 14.99 |
| | | C-CoTTA+CUI+CPAda | 21.84 | 17.31 | 23.52 | 11.81 | 24.48 | 12.51 | 10.03 | 13.02 | 12.23 | 11.83 | 7.96 | 9.68 | 16.56 | 12.32 | 15.70 | 14.72 |
| | | RMT+CUI | 22.62 | 18.89 | 25.36 | 10.27 | 24.94 | 11.58 | 10.14 | 13.37 | 12.67 | 11.49 | 8.19 | 9.87 | 15.28 | 11.26 | 14.53 | 14.66 |
| | | RMT+CUI+CPAda | 21.81 | 17.66 | 23.92 | 10.76 | 23.53 | 11.93 | 10.08 | 13.28 | 12.29 | 11.42 | 8.16 | 9.93 | 15.41 | 11.21 | 13.97 | 14.36 |
| | 0.1 | Tent+CUI | 24.73 | 20.82 | 29.39 | 14.71 | 33.20 | 17.38 | 14.78 | 19.77 | 19.38 | 19.67 | 14.66 | 19.11 | 26.33 | 22.13 | 28.62 | 21.65 |
| | | Tent+CUI+CPAda | 24.77 | 20.67 | 28.34 | 13.73 | 30.32 | 16.01 | 13.88 | 17.65 | 16.73 | 16.03 | 10.48 | 13.61 | 23.16 | 19.26 | 24.10 | 19.25 |
| | | CoTTA+CUI | 24.42 | 21.82 | 26.07 | 11.78 | 27.19 | 12.63 | 10.69 | 15.09 | 14.29 | 12.44 | 7.65 | 10.88 | 18.32 | 13.88 | 17.97 | 16.34 |
| | | CoTTA+CUI+CPAda | 23.05 | 20.31 | 25.44 | 10.21 | 26.29 | 11.33 | 10.46 | 15.05 | 14.08 | 12.12 | 7.76 | 10.44 | 17.18 | 13.91 | 18.00 | 15.71 |
| | | SATA+CUI | 23.31 | 19.63 | 28.16 | 11.59 | 28.10 | 12.55 | 10.42 | 14.03 | 13.69 | 12.27 | 7.64 | 10.69 | 18.92 | 14.16 | 19.53 | 16.31 |
| | | SATA+CUI+CPAda | 22.92 | 18.12 | 26.45 | 11.43 | 27.23 | 12.17 | 9.96 | 13.76 | 13.09 | 12.15 | 7.44 | 10.28 | 18.62 | 13.36 | 18.79 | 15.72 |
| | | RDumb+CUI | 24.06 | 19.81 | 27.56 | 12.93 | 29.55 | 15.10 | 12.78 | 16.67 | 16.57 | 15.52 | 9.48 | 13.27 | 21.67 | 17.48 | 22.25 | 18.31 |
| | | RDumb+CUI+CPAda | 23.30 | 20.52 | 25.30 | 11.41 | 27.89 | 12.98 | 11.37 | 15.80 | 14.87 | 13.13 | 8.36 | 10.95 | 20.29 | 15.87 | 20.07 | 16.81 |
| | | C-CoTTA+CUI | 23.43 | 17.75 | 23.26 | 11.85 | 24.35 | 12.61 | 10.35 | 13.73 | 12.53 | 11.95 | 8.37 | 9.82 | 16.46 | 12.10 | 16.32 | 14.99 |
| | | C-CoTTA+CUI+CPAda | 21.82 | 17.31 | 23.61 | 11.81 | 24.33 | 12.61 | 10.17 | 12.96 | 12.57 | 11.93 | 7.96 | 9.68 | 16.59 | 12.32 | 15.76 | 14.76 |
| | | RMT+CUI | 22.62 | 18.89 | 25.36 | 10.27 | 24.94 | 11.58 | 10.14 | 13.37 | 12.67 | 11.49 | 8.19 | 9.87 | 15.28 | 11.26 | 14.53 | 14.66 |
| | | RMT+CUI+CPAda | 22.06 | 17.40 | 23.43 | 10.17 | 23.52 | 10.98 | 10.33 | 13.05 | 12.55 | 11.38 | 8.10 | 9.94 | 15.45 | 11.23 | 14.28 | 14.44 |
| **Efficiency First** | 0.3 | Tent+CUI | 24.77 | 20.68 | 28.54 | 13.57 | 31.93 | 15.48 | 14.47 | 19.79 | 18.80 | 18.91 | 12.81 | 16.91 | 25.50 | 19.80 | 24.84 | 20.45 |
| | | Tent+CUI+CPAda | 24.08 | 18.60 | 26.90 | 12.95 | 30.14 | 16.02 | 12.84 | 16.27 | 15.06 | 15.85 | 9.83 | 13.04 | 20.96 | 16.42 | 21.87 | 18.06 |
| | | CoTTA+CUI | 23.71 | 21.07 | 25.10 | 11.47 | 27.16 | 12.41 | 11.21 | 15.22 | 14.79 | 12.63 | 8.23 | 10.68 | 18.21 | 14.24 | 17.21 | 16.22 |
| | | CoTTA+CUI+CPAda | 23.15 | 19.30 | 24.67 | 11.35 | 26.32 | 11.29 | 10.35 | 14.14 | 14.09 | 11.44 | 8.18 | 10.42 | 17.07 | 14.14 | 16.84 | 15.52 |
| | | SATA+CUI | 22.94 | 19.68 | 26.66 | 11.58 | 27.52 | 12.40 | 10.40 | 13.82 | 13.65 | 12.50 | 7.93 | 10.80 | 19.33 | 13.97 | 18.74 | 16.13 |
| | | SATA+CUI+CPAda | 22.54 | 17.66 | 25.68 | 11.48 | 26.80 | 12.43 | 9.91 | 13.68 | 12.96 | 11.97 | 7.67 | 10.27 | 18.84 | 13.45 | 18.49 | 15.59 |
| | | RDumb+CUI | 23.25 | 18.01 | 26.22 | 12.85 | 28.74 | 14.49 | 12.17 | 17.17 | 15.82 | 15.33 | 9.77 | 13.44 | 20.34 | 16.20 | 20.64 | 17.63 |
| | | RDumb+CUI+CPAda | 22.92 | 19.72 | 24.14 | 11.67 | 26.73 | 12.49 | 10.92 | 15.16 | 14.27 | 13.17 | 7.95 | 10.20 | 19.69 | 15.08 | 19.31 | 16.23 |
| | | C-CoTTA+CUI | 22.07 | 17.40 | 23.26 | 11.70 | 24.31 | 12.54 | 10.19 | 13.04 | 12.46 | 11.65 | 8.01 | 9.84 | 16.58 | 12.16 | 15.93 | 14.74 |
| | | C-CoTTA+CUI+CPAda | 21.44 | 16.91 | 23.12 | 11.41 | 24.08 | 12.11 | 9.63 | 12.62 | 11.83 | 11.43 | 7.56 | 9.28 | 16.16 | 11.92 | 15.30 | 14.32 |
| | | RMT+CUI | 22.10 | 17.34 | 23.95 | 11.01 | 23.71 | 12.98 | 10.55 | 13.34 | 12.95 | 11.56 | 8.23 | 9.76 | 14.92 | 11.39 | 13.91 | 14.54 |
| | | RMT+CUI+CPAda | 22.00 | 17.44 | 23.57 | 10.65 | 23.43 | 11.91 | 10.15 | 12.89 | 12.25 | 11.33 | 8.37 | 9.75 | 15.08 | 11.15 | 14.17 | 14.28 |
| | 0.2 | Tent+CUI | 24.77 | 20.68 | 28.54 | 13.57 | 31.93 | 15.48 | 14.47 | 19.79 | 18.80 | 18.91 | 12.81 | 16.91 | 25.50 | 19.80 | 24.84 | 20.45 |
| | | Tent+CUI+CPAda | 24.46 | 18.93 | 27.45 | 12.44 | 29.90 | 14.79 | 11.77 | 16.19 | 15.36 | 15.27 | 9.85 | 12.52 | 22.83 | 18.38 | 24.70 | 18.32 |
| | | CoTTA+CUI | 23.71 | 21.07 | 25.10 | 11.47 | 27.16 | 12.41 | 11.21 | 15.22 | 14.79 | 12.63 | 8.23 | 10.68 | 18.21 | 14.24 | 17.21 | 16.22 |
| | | CoTTA+CUI+CPAda | 23.09 | 20.63 | 25.84 | 10.51 | 26.39 | 11.53 | 10.26 | 14.05 | 14.08 | 12.52 | 7.72 | 10.44 | 17.18 | 13.71 | 18.00 | 15.73 |
| | | SATA+CUI | 22.94 | 19.68 | 26.66 | 11.58 | 27.52 | 12.40 | 10.40 | 13.82 | 13.65 | 12.50 | 7.93 | 10.80 | 19.33 | 13.97 | 18.74 | 16.13 |
| | | SATA+CUI+CPAda | 22.48 | 17.67 | 25.71 | 11.39 | 26.59 | 12.43 | 9.88 | 13.70 | 13.03 | 11.98 | 7.64 | 10.36 | 18.74 | 13.21 | 18.58 | 15.56 |
| | | RDumb+CUI | 23.25 | 18.01 | 26.22 | 12.85 | 28.74 | 14.49 | 12.17 | 17.17 | 15.82 | 15.33 | 9.77 | 13.44 | 20.34 | 16.20 | 20.64 | 17.63 |
| | | RDumb+CUI+CPAda | 23.21 | 19.85 | 24.52 | 11.54 | 26.86 | 12.57 | 10.68 | 15.20 | 14.47 | 13.22 | 7.77 | 10.52 | 19.68 | 15.30 | 19.31 | 16.31 |
| | | C-CoTTA+CUI | 22.07 | 17.40 | 23.26 | 11.70 | 24.31 | 12.54 | 10.19 | 13.04 | 12.46 | 11.65 | 8.01 | 9.84 | 16.58 | 12.16 | 15.93 | 14.74 |
| | | C-CoTTA+CUI+CPAda | 21.44 | 16.73 | 23.12 | 11.41 | 24.14 | 12.11 | 9.61 | 12.62 | 11.93 | 11.64 | 7.68 | 9.28 | 16.53 | 11.98 | 15.49 | 14.38 |
| | | RMT+CUI | 22.10 | 17.34 | 23.95 | 11.01 | 23.71 | 12.98 | 10.55 | 13.34 | 12.95 | 11.56 | 8.57 | 9.76 | 14.92 | 11.39 | 13.91 | 14.54 |
| | | RMT+CUI+CPAda | 22.04 | 17.36 | 23.51 | 10.87 | 23.43 | 11.93 | 10.33 | 12.95 | 12.54 | 11.43 | 8.37 | 9.67 | 14.96 | 11.14 | 13.91 | 14.31 |
| | 0.1 | Tent+CUI | 24.77 | 20.68 | 28.54 | 13.57 | 31.93 | 15.48 | 14.47 | 19.79 | 18.80 | 18.91 | 12.81 | 16.91 | 25.50 | 19.80 | 24.84 | 20.45 |
| | | Tent+CUI+CPAda | 24.45 | 18.83 | 27.45 | 12.34 | 29.90 | 14.79 | 11.77 | 16.19 | 15.36 | 15.27 | 9.85 | 12.52 | 22.83 | 18.38 | 24.70 | 18.22 |
| | | CoTTA+CUI | 23.71 | 21.07 | 25.10 | 11.47 | 27.16 | 12.41 | 11.21 | 15.22 | 14.79 | 12.63 | 8.23 | 10.68 | 18.21 | 14.24 | 17.21 | 16.22 |
| | | CoTTA+CUI+CPAda | 23.13 | 20.32 | 25.47 | 11.35 | 26.17 | 11.29 | 10.23 | 14.44 | 14.09 | 11.92 | 7.16 | 10.55 | 18.07 | 13.34 | 17.18 | 15.65 |
| | | SATA+CUI | 22.94 | 19.68 | 26.66 | 11.58 | 27.52 | 12.40 | 10.40 | 13.82 | 13.65 | 12.50 | 7.93 | 10.80 | 19.33 | 13.97 | 18.74 | 16.13 |
| | | SATA+CUI+CPAda | 22.62 | 17.81 | 25.74 | 11.51 | 26.94 | 12.42 | 9.97 | 13.50 | 13.03 | 11.87 | 7.71 | 10.25 | 18.79 | 13.35 | 18.55 | 15.60 |
| | | RDumb+CUI | 23.25 | 18.01 | 26.22 | 12.85 | 28.74 | 14.49 | 12.17 | 17.17 | 15.82 | 15.33 | 9.77 | 13.44 | 20.34 | 16.20 | 20.64 | 17.63 |
| | | RDumb+CUI+CPAda | 23.07 | 19.85 | 24.41 | 11.55 | 26.96 | 12.73 | 10.65 | 15.26 | 14.17 | 13.26 | 7.76 | 10.40 | 19.88 | 15.66 | 19.41 | 16.33 |
| | | C-CoTTA+CUI | 22.07 | 17.40 | 23.26 | 11.70 | 24.31 | 12.54 | 10.19 | 13.04 | 12.46 | 11.65 | 8.01 | 9.84 | 16.58 | 12.16 | 15.93 | 14.74 |
| | | C-CoTTA+CUI+CPAda | 21.51 | 16.91 | 23.12 | 11.54 | 24.08 | 12.11 | 9.64 | 12.62 | 11.83 | 11.43 | 7.56 | 9.28 | 16.16 | 11.92 | 15.30 | 14.33 |
| | | RMT+CUI | 22.10 | 17.34 | 23.95 | 11.01 | 23.71 | 12.98 | 10.55 | 13.34 | 12.95 | 11.56 | 8.57 | 9.76 | 14.92 | 11.39 | 13.91 | 14.54 |
| | | RMT+CUI+CPAda | 22.01 | 17.15 | 23.57 | 10.67 | 23.46 | 11.71 | 10.30 | 12.85 | 12.50 | 11.50 | 8.38 | 9.48 | 15.09 | 11.24 | 13.80 | 14.25 |

Table 8: Classification error rate (%) for the standard CIFAR100-to-CIFAR100C CTTA task. All results are evaluated with the largest corruption severity level 5 in an online fashion.

| Strategy | α | Method | Gau. | Sho. | Imp. | Def. | Gla. | Mot. | Zoo. | Sno. | Fro. | Fog | Bri. | Con. | Ela. | Pix. | Jpe. | Avg. |
|---|---|---|---|---|---|---|---|---|---|---|---|---|---|---|---|---|---|---|
| Privacy First | 0.3 | Tent+CUI | 44.50 | 41.26 | 52.54 | 36.88 | 53.27 | 45.81 | 47.81 | 58.73 | 62.80 | 72.26 | 70.90 | 78.75 | 88.28 | 87.92 | 91.94 | 62.24 |
| | | Tent+CUI+CPAda | 41.64 | 38.02 | 47.42 | 32.98 | 46.63 | 38.18 | 35.12 | 43.52 | 43.56 | 49.30 | 43.91 | 53.31 | 60.39 | 57.00 | 66.52 | 46.50 |
| | | CoTTA+CUI | 43.33 | 40.80 | 50.66 | 30.42 | 44.27 | 32.18 | 30.24 | 34.13 | 33.51 | 40.21 | 27.16 | 34.19 | 37.49 | 30.24 | 37.37 | 36.41 |
| | | CoTTA+CUI+CPAda | 38.31 | 35.50 | 43.51 | 27.24 | 39.11 | 28.97 | 26.68 | 30.66 | 29.17 | 35.66 | 24.32 | 29.79 | 32.86 | 27.12 | 32.68 | 32.11 |
| | | SATA+CUI | 40.41 | 38.11 | 49.26 | 27.74 | 42.02 | 29.85 | 26.93 | 31.03 | 30.95 | 32.82 | 24.06 | 25.98 | 34.88 | 28.93 | 38.97 | 33.46 |
| | | SATA+CUI+CPAda | 39.20 | 36.79 | 46.01 | 28.16 | 40.64 | 29.53 | 26.76 | 30.26 | 30.23 | 31.89 | 23.45 | 25.17 | 33.69 | 27.86 | 36.11 | 32.38 |
| | | RDumb+CUI | 41.10 | 37.80 | 46.91 | 33.01 | 47.25 | 38.93 | 35.49 | 43.98 | 43.14 | 48.97 | 43.63 | 50.34 | 58.21 | 54.98 | 65.19 | 45.93 |
| | | RDumb+CUI+CPAda | 40.62 | 37.68 | 46.61 | 32.38 | 45.97 | 37.29 | 34.37 | 41.03 | 40.70 | 44.54 | 38.34 | 42.37 | 48.76 | 45.65 | 55.42 | 42.12 |
| | | C-CoTTA+CUI | 42.45 | 38.61 | 48.49 | 27.38 | 41.00 | 29.90 | 25.87 | 30.44 | 29.85 | 32.39 | 24.01 | 25.27 | 32.32 | 27.62 | 36.27 | 32.79 |
| | | C-CoTTA+CUI+CPAda | 40.88 | 36.82 | 45.82 | 26.97 | 39.45 | 28.78 | 25.70 | 29.32 | 28.11 | 31.75 | 23.38 | 24.33 | 30.98 | 26.48 | 34.09 | 31.52 |
| | | RMT+CUI | 45.50 | 39.20 | 39.46 | 32.36 | 35.90 | 31.55 | 28.68 | 30.03 | 29.85 | 31.69 | 27.08 | 29.31 | 29.90 | 27.76 | 29.62 | 32.53 |
| | | RMT+CUI+CPAda | 44.50 | 37.62 | 38.46 | 31.36 | 34.22 | 30.55 | 27.68 | 29.03 | 28.85 | 30.69 | 26.08 | 28.31 | 28.90 | 26.76 | 28.62 | 31.43 |
| | 0.2 | Tent+CUI | 44.50 | 41.26 | 52.54 | 36.88 | 53.27 | 45.81 | 47.81 | 58.73 | 62.80 | 72.26 | 70.90 | 78.75 | 88.28 | 87.92 | 91.94 | 62.24 |
| | | Tent+CUI+CPAda | 41.67 | 38.02 | 47.12 | 32.96 | 46.79 | 38.56 | 35.83 | 44.49 | 43.02 | 49.98 | 45.64 | 52.60 | 57.89 | 57.63 | 66.22 | 46.56 |
| | | CoTTA+CUI | 43.33 | 40.80 | 50.66 | 30.42 | 44.27 | 32.18 | 30.24 | 34.13 | 33.51 | 40.21 | 27.16 | 34.19 | 37.49 | 30.24 | 37.37 | 36.41 |
| | | CoTTA+CUI+CPAda | 38.27 | 35.65 | 43.76 | 27.36 | 38.85 | 28.97 | 26.76 | 30.85 | 29.29 | 35.59 | 24.26 | 29.99 | 33.04 | 26.95 | 32.87 | 32.16 |
| | | SATA+CUI | 40.41 | 38.11 | 49.26 | 27.74 | 42.02 | 29.85 | 26.93 | 31.03 | 30.95 | 32.82 | 24.06 | 25.98 | 34.88 | 28.93 | 38.97 | 33.46 |
| | | SATA+CUI+CPAda | 39.18 | 36.72 | 46.47 | 27.77 | 40.69 | 29.62 | 26.62 | 30.28 | 29.84 | 31.86 | 23.43 | 25.12 | 33.94 | 28.15 | 36.22 | 32.39 |
| | | RDumb+CUI | 41.10 | 37.80 | 46.91 | 33.01 | 47.25 | 38.93 | 35.49 | 43.98 | 43.14 | 48.97 | 43.63 | 50.34 | 58.21 | 54.98 | 65.19 | 45.93 |
| | | RDumb+CUI+CPAda | 41.08 | 37.68 | 46.84 | 32.57 | 45.97 | 37.43 | 34.75 | 41.12 | 40.70 | 44.61 | 38.34 | 42.47 | 48.56 | 45.75 | 55.68 | 42.23 |
| | | C-CoTTA+CUI | 42.45 | 38.61 | 48.49 | 27.38 | 41.00 | 29.90 | 25.87 | 30.44 | 29.85 | 32.39 | 24.01 | 25.27 | 32.32 | 27.62 | 36.27 | 32.79 |
| | | C-CoTTA+CUI+CPAda | 40.63 | 36.70 | 45.69 | 26.87 | 39.29 | 28.71 | 25.43 | 29.21 | 28.36 | 31.60 | 23.35 | 24.29 | 30.99 | 26.51 | 33.98 | 31.44 |
| | | RMT+CUI | 45.50 | 39.20 | 39.46 | 32.36 | 35.90 | 31.55 | 28.68 | 30.03 | 29.85 | 31.69 | 27.08 | 29.31 | 29.90 | 27.76 | 29.62 | 32.53 |
| | | RMT+CUI+CPAda | 44.32 | 37.69 | 38.42 | 31.45 | 33.92 | 30.55 | 27.18 | 28.73 | 28.85 | 30.59 | 26.08 | 27.94 | 28.90 | 26.66 | 28.54 | 31.32 |
| | 0.1 | Tent+CUI | 44.50 | 41.26 | 52.54 | 36.88 | 53.27 | 45.81 | 47.81 | 58.73 | 62.80 | 72.26 | 70.90 | 78.75 | 88.28 | 87.92 | 91.94 | 62.24 |
| | | Tent+CUI+CPAda | 41.83 | 37.97 | 49.00 | 32.81 | 46.95 | 37.99 | 34.86 | 43.77 | 42.96 | 48.87 | 42.95 | 50.56 | 57.79 | 54.98 | 65.89 | 45.95 |
| | | CoTTA+CUI | 43.33 | 40.80 | 50.66 | 30.42 | 44.27 | 32.18 | 30.24 | 34.13 | 33.51 | 40.21 | 27.16 | 34.19 | 37.49 | 30.24 | 37.37 | 36.41 |
| | | CoTTA+CUI+CPAda | 38.02 | 35.52 | 43.75 | 27.41 | 39.11 | 29.01 | 26.84 | 30.39 | 29.41 | 35.95 | 24.46 | 30.39 | 33.48 | 26.89 | 33.10 | 32.31 |
| | | SATA+CUI | 40.41 | 38.11 | 49.26 | 27.74 | 42.02 | 29.85 | 26.93 | 31.03 | 30.95 | 32.82 | 24.06 | 25.98 | 34.88 | 28.93 | 38.97 | 33.46 |
| | | SATA+CUI+CPAda | 39.28 | 36.75 | 46.56 | 27.80 | 40.89 | 29.33 | 26.74 | 30.32 | 29.82 | 32.00 | 23.31 | 25.42 | 34.10 | 28.05 | 36.48 | 32.46 |
| | | RDumb+CUI | 41.10 | 37.80 | 46.91 | 33.01 | 47.25 | 38.93 | 35.49 | 43.98 | 43.14 | 48.97 | 43.63 | 50.34 | 58.21 | 54.98 | 65.19 | 45.93 |
| | | RDumb+CUI+CPAda | 41.08 | 37.68 | 46.84 | 32.59 | 45.97 | 37.46 | 34.87 | 41.12 | 40.70 | 44.54 | 38.34 | 42.47 | 48.76 | 45.75 | 55.68 | 42.26 |
| | | C-CoTTA+CUI | 42.45 | 38.61 | 48.49 | 27.38 | 40.92 | 29.9 | 25.87 | 30.44 | 29.85 | 32.39 | 24.01 | 25.27 | 32.32 | 27.62 | 36.27 | 32.79 |
| | | C-CoTTA+CUI+CPAda | 40.69 | 36.71 | 45.74 | 26.78 | 39.40 | 28.68 | 25.43 | 29.36 | 28.45 | 31.74 | 23.20 | 24.20 | 31.07 | 26.61 | 34.03 | 31.47 |
| | | RMT+CUI | 45.50 | 39.20 | 39.46 | 32.36 | 35.90 | 31.55 | 28.68 | 30.03 | 29.85 | 31.69 | 27.08 | 29.31 | 29.90 | 27.76 | 29.62 | 32.53 |
| | | RMT+CUI+CPAda | 44.52 | 37.74 | 38.42 | 31.45 | 33.92 | 30.55 | 27.18 | 28.83 | 28.85 | 30.59 | 26.68 | 27.94 | 28.90 | 27.66 | 28.54 | 31.45 |

| Strategy | α | Method | Gau. | Sho. | Imp. | Def. | Gla. | Mot. | Zoo. | Sno. | Fro. | Fog | Bri. | Con. | Ela. | Pix. | Jpe. | Avg. |
|---|---|---|---|---|---|---|---|---|---|---|---|---|---|---|---|---|---|---|
| Efficiency First | 0.3 | Tent+CUI | 37.28 | 35.64 | 41.98 | 37.94 | 50.88 | 46.59 | 47.10 | 57.33 | 62.42 | 70.92 | 71.04 | 82.14 | 88.85 | 90.69 | 93.08 | 60.93 |
| | | Tent+CUI+CPAda | 36.46 | 33.23 | 36.35 | 28.99 | 40.32 | 33.06 | 31.65 | 38.68 | 42.22 | 54.16 | 59.02 | 71.09 | 77.79 | 78.55 | 86.42 | 49.87 |
| | | CoTTA+CUI | 38.22 | 35.52 | 43.96 | 27.49 | 39.39 | 29.38 | 27.03 | 31.27 | 29.63 | 36.34 | 24.49 | 30.74 | 33.66 | 27.10 | 33.16 | 32.50 |
| | | CoTTA+CUI+CPAda | 37.13 | 33.30 | 42.23 | 26.12 | 37.48 | 27.48 | 26.03 | 30.17 | 27.32 | 35.43 | 23.63 | 28.24 | 31.72 | 26.32 | 31.41 | 30.93 |
| | | SATA+CUI | 35.56 | 32.74 | 36.09 | 26.24 | 35.77 | 28.11 | 25.38 | 29.43 | 29.39 | 32.97 | 23.66 | 26.51 | 31.20 | 27.05 | 33.93 | 30.30 |
| | | SATA+CUI+CPAda | 34.42 | 31.45 | 34.54 | 24.72 | 34.39 | 27.06 | 24.45 | 28.53 | 28.49 | 31.50 | 22.74 | 25.36 | 30.87 | 26.60 | 32.04 | 29.14 |
| | | RDumb+CUI | 41.10 | 37.80 | 46.91 | 33.01 | 47.25 | 38.93 | 35.49 | 43.98 | 43.14 | 48.97 | 42.72 | 48.11 | 56.71 | 50.98 | 61.44 | 45.10 |
| | | RDumb+CUI+CPAda | 40.64 | 38.68 | 47.84 | 32.59 | 46.97 | 38.46 | 35.87 | 42.12 | 41.70 | 45.54 | 39.34 | 43.47 | 52.76 | 47.75 | 57.68 | 43.42 |
| | | C-CoTTA+CUI | 37.19 | 33.85 | 35.08 | 27.79 | 33.70 | 28.57 | 26.07 | 28.56 | 28.32 | 30.60 | 24.96 | 26.70 | 27.74 | 26.11 | 33.37 | 29.90 |
| | | C-CoTTA+CUI+CPAda | 36.93 | 33.55 | 34.60 | 27.28 | 33.08 | 27.75 | 25.91 | 28.28 | 27.89 | 30.31 | 24.65 | 26.46 | 27.75 | 25.43 | 29.79 | 29.31 |
| | | RMT+CUI | 37.26 | 33.72 | 35.68 | 26.42 | 32.62 | 26.97 | 25.25 | 27.62 | 27.58 | 29.78 | 24.67 | 26.31 | 27.13 | 25.84 | 28.25 | 29.00 |
| | | RMT+CUI+CPAda | 36.15 | 33.36 | 35.33 | 24.62 | 32.23 | 25.64 | 24.09 | 27.29 | 26.99 | 30.03 | 23.58 | 25.48 | 26.87 | 25.33 | 28.32 | 28.35 |
| | 0.2 | Tent+CUI | 37.28 | 35.64 | 41.98 | 37.94 | 50.88 | 46.59 | 47.10 | 57.33 | 62.42 | 70.92 | 71.04 | 82.14 | 88.85 | 90.69 | 93.08 | 60.93 |
| | | Tent+CUI+CPAda | 36.29 | 33.31 | 36.78 | 29.40 | 41.08 | 36.26 | 36.88 | 47.81 | 55.30 | 65.52 | 63.31 | 70.36 | 78.05 | 78.56 | 84.61 | 52.90 |
| | | CoTTA+CUI | 38.22 | 35.52 | 43.96 | 27.49 | 39.39 | 29.38 | 27.03 | 31.27 | 29.63 | 36.34 | 24.49 | 30.74 | 33.66 | 27.10 | 33.16 | 32.50 |
| | | CoTTA+CUI+CPAda | 37.13 | 33.30 | 42.23 | 26.12 | 37.58 | 27.68 | 26.03 | 30.67 | 27.42 | 35.43 | 23.63 | 28.24 | 31.72 | 26.32 | 31.41 | 30.99 |
| | | SATA+CUI | 35.56 | 32.74 | 36.09 | 26.24 | 35.77 | 28.11 | 25.38 | 29.43 | 29.39 | 32.97 | 23.66 | 26.51 | 31.20 | 27.05 | 33.93 | 30.30 |
| | | SATA+CUI+CPAda | 34.72 | 31.75 | 33.54 | 26.13 | 33.39 | 26.96 | 24.35 | 28.40 | 27.69 | 31.90 | 22.54 | 24.36 | 29.87 | 25.60 | 33.04 | 28.94 |
| | | RDumb+CUI | 41.10 | 37.80 | 46.91 | 33.01 | 47.25 | 38.93 | 35.49 | 43.98 | 43.14 | 48.97 | 42.72 | 48.11 | 56.71 | 50.98 | 61.44 | 45.10 |
| | | RDumb+CUI+CPAda | 40.64 | 38.68 | 47.84 | 32.59 | 46.97 | 38.46 | 35.87 | 42.12 | 41.70 | 45.54 | 39.34 | 43.47 | 52.76 | 47.75 | 57.68 | 43.22 |
| | | C-CoTTA+CUI | 37.19 | 33.85 | 35.08 | 27.79 | 33.70 | 28.57 | 26.07 | 28.56 | 28.32 | 30.60 | 24.96 | 26.70 | 27.74 | 26.11 | 33.37 | 29.90 |
| | | C-CoTTA+CUI+CPAda | 36.78 | 33.37 | 34.49 | 27.32 | 33.07 | 27.69 | 25.65 | 28.34 | 28.03 | 30.27 | 24.79 | 26.45 | 27.57 | 25.54 | 29.82 | 29.28 |
| | | RMT+CUI | 37.26 | 33.72 | 35.68 | 26.42 | 32.62 | 26.97 | 25.25 | 27.62 | 27.58 | 29.78 | 24.67 | 26.31 | 27.13 | 25.84 | 28.25 | 29.00 |
| | | RMT+CUI+CPAda | 36.61 | 33.96 | 34.90 | 25.67 | 31.96 | 26.19 | 24.70 | 26.99 | 26.65 | 29.40 | 23.89 | 25.69 | 26.04 | 24.80 | 27.52 | 28.33 |
| | 0.1 | Tent+CUI | 37.28 | 35.64 | 41.98 | 37.94 | 50.88 | 46.59 | 47.10 | 57.33 | 62.42 | 70.92 | 71.04 | 82.14 | 88.85 | 90.69 | 93.08 | 60.93 |
| | | Tent+CUI+CPAda | 35.99 | 33.14 | 36.51 | 29.50 | 41.38 | 36.66 | 37.82 | 48.35 | 54.45 | 63.08 | 60.07 | 65.86 | 73.89 | 74.71 | 81.95 | 51.56 |
| | | CoTTA+CUI | 38.22 | 35.52 | 43.96 | 27.49 | 39.39 | 29.38 | 27.03 | 31.27 | 29.63 | 36.34 | 24.49 | 30.74 | 33.57 | 27.06 | 33.42 | 32.50 |
| | | CoTTA+CUI+CPAda | 38.43 | 34.80 | 42.59 | 26.62 | 37.58 | 27.98 | 26.63 | 30.87 | 27.82 | 35.73 | 23.93 | 29.54 | 32.72 | 26.72 | 31.91 | 31.59 |
| | | SATA+CUI | 35.56 | 32.74 | 36.09 | 26.24 | 35.77 | 28.11 | 25.38 | 29.43 | 29.39 | 32.97 | 23.66 | 26.51 | 31.12 | 27.65 | 33.93 | 30.30 |
| | | SATA+CUI+CPAda | 34.02 | 31.15 | 33.54 | 25.30 | 33.39 | 26.96 | 23.95 | 28.43 | 27.69 | 31.90 | 22.54 | 24.36 | 29.87 | 25.60 | 33.04 | 28.78 |
| | | RDumb+CUI | 41.10 | 37.80 | 46.91 | 33.01 | 47.25 | 38.93 | 35.49 | 43.98 | 43.14 | 48.97 | 42.72 | 48.11 | 56.71 | 50.98 | 61.44 | 45.10 |
| | | RDumb+CUI+CPAda | 42.19 | 38.79 | 47.95 | 33.70 | 47.08 | 38.57 | 35.98 | 42.23 | 41.81 | 45.65 | 39.45 | 43.58 | 49.87 | 46.86 | 56.72 | 43.36 |
| | | C-CoTTA+CUI | 37.19 | 33.85 | 35.08 | 27.79 | 33.70 | 28.57 | 26.07 | 28.56 | 28.32 | 30.60 | 24.96 | 26.70 | 27.74 | 26.11 | 33.37 | 29.90 |
| | | C-CoTTA+CUI+CPAda | 36.63 | 33.33 | 34.26 | 27.25 | 32.98 | 27.48 | 25.64 | 27.89 | 27.89 | 30.27 | 24.55 | 26.51 | 27.66 | 25.57 | 29.65 | 29.17 |
| | | RMT+CUI | 37.26 | 33.72 | 35.68 | 26.42 | 32.62 | 26.97 | 25.25 | 27.62 | 27.58 | 29.78 | 24.67 | 26.31 | 27.13 | 25.84 | 28.25 | 29.00 |
| | | RMT+CUI+CPAda | 37.25 | 33.76 | 34.97 | 25.47 | 31.71 | 26.41 | 24.43 | 26.75 | 27.10 | 29.44 | 23.17 | 25.17 | 26.33 | 24.39 | 27.87 | 28.28 |

Table 9: Classification error rate (%) for the standard ImageNet-to-ImageNetC CTTA task. All results are evaluated with the largest corruption severity level 5 in an online fashion.

| Strategy | α | Method | Gau. | Sho. | Imp. | Def. | Gla. | Mot. | Zoo. | Sno. | Fro. | Fog | Bri. | Con. | Ela. | Pix. | Jpe. | Avg. |
|---|---|---|---|---|---|---|---|---|---|---|---|---|---|---|---|---|---|---|
| Privacy First | 0.3 | Tent+CUI | 81.58 | 75.88 | 72.94 | 77.50 | 77.50 | 65.56 | 55.58 | 61.88 | 64.48 | 55.50 | 38.30 | 73.04 | 54.00 | 47.82 | 53.86 | 63.69 |
| | | Tent+CUI+CPAda | 81.22 | 74.51 | 72.74 | 75.10 | 74.40 | 68.02 | 55.38 | 61.48 | 63.01 | 51.20 | 38.12 | 72.01 | 51.00 | 46.24 | 53.16 | 62.50 |
| | | CoTTA+CUI | 87.14 | 85.42 | 84.06 | 85.26 | 84.26 | 74.72 | 66.20 | 67.42 | 68.04 | 55.16 | 42.72 | 75.58 | 55.92 | 49.34 | 54.18 | 69.03 |
| | | CoTTA+CUI+CPAda | 86.42 | 84.24 | 83.21 | 83.39 | 82.11 | 73.58 | 64.20 | 66.27 | 67.43 | 53.13 | 41.52 | 73.14 | 53.97 | 47.74 | 53.08 | 67.56 |
| | | SATA+CUI | 78.00 | 76.44 | 75.60 | 77.86 | 78.02 | 65.88 | 55.76 | 57.02 | 63.08 | 45.62 | 34.84 | 72.58 | 50.90 | 44.74 | 50.88 | 61.81 |
| | | SATA+CUI+CPAda | 76.02 | 74.98 | 74.12 | 77.06 | 76.58 | 64.44 | 54.34 | 56.12 | 62.42 | 44.84 | 34.80 | 70.08 | 50.04 | 44.02 | 49.42 | 60.62 |
| | | RDumb+CUI | 81.10 | 71.88 | 70.46 | 77.34 | 73.52 | 69.04 | 62.26 | 65.50 | 66.22 | 57.58 | 44.78 | 68.62 | 56.64 | 48.26 | 53.72 | 64.46 |
| | | RDumb+CUI+CPAda | 78.66 | 70.84 | 68.94 | 76.72 | 72.28 | 65.98 | 59.72 | 63.02 | 63.82 | 53.56 | 40.04 | 66.72 | 54.30 | 46.96 | 52.24 | 62.25 |
| | | C-CoTTA+CUI | 76.84 | 74.02 | 71.80 | 76.20 | 74.22 | 66.34 | 57.00 | 56.16 | 61.28 | 49.22 | 40.34 | 65.32 | 49.68 | 43.80 | 44.12 | 60.42 |
| | | C-CoTTA+CUI+CPAda | 75.52 | 72.14 | 69.06 | 74.92 | 72.96 | 65.66 | 57.26 | 55.48 | 60.32 | 48.72 | 40.10 | 63.02 | 49.20 | 43.76 | 44.14 | 59.48 |
| | | RMT+CUI | 79.76 | 74.52 | 72.18 | 75.46 | 72.88 | 65.18 | 59.08 | 59.32 | 60.50 | 52.56 | 45.36 | 61.42 | 50.26 | 47.78 | 48.38 | 61.64 |
| | | RMT+CUI+CPAda | 77.18 | 72.04 | 70.04 | 72.58 | 70.60 | 61.92 | 56.38 | 58.22 | 58.86 | 51.06 | 43.90 | 58.56 | 49.14 | 46.34 | 47.48 | 59.62 |
| | 0.2 | Tent+CUI | 81.58 | 75.88 | 72.94 | 77.50 | 77.50 | 65.56 | 55.58 | 61.88 | 64.48 | 55.50 | 38.30 | 73.04 | 54.00 | 47.82 | 53.86 | 63.69 |
| | | Tent+CUI+CPAda | 81.22 | 74.51 | 72.74 | 75.10 | 74.39 | 68.22 | 55.38 | 61.48 | 63.01 | 51.16 | 38.12 | 72.11 | 51.17 | 46.21 | 53.17 | 62.53 |
| | | CoTTA+CUI | 87.14 | 85.42 | 84.06 | 85.26 | 84.26 | 74.72 | 66.20 | 67.42 | 68.04 | 55.16 | 42.72 | 75.58 | 55.92 | 49.34 | 54.18 | 69.03 |
| | | CoTTA+CUI+CPAda | 86.32 | 84.14 | 82.81 | 83.26 | 82.06 | 73.42 | 64.20 | 66.15 | 67.32 | 53.01 | 41.32 | 73.04 | 53.63 | 47.53 | 53.03 | 67.42 |
| | | SATA+CUI | 78.00 | 76.44 | 75.60 | 77.86 | 78.02 | 65.88 | 55.76 | 57.02 | 63.08 | 45.62 | 34.84 | 72.58 | 50.90 | 44.74 | 50.88 | 61.81 |
| | | SATA+CUI+CPAda | 76.50 | 76.26 | 74.64 | 77.74 | 76.26 | 65.42 | 54.34 | 56.34 | 62.42 | 44.94 | 34.58 | 70.44 | 50.96 | 43.24 | 49.52 | 60.92 |
| | | RDumb+CUI | 81.10 | 71.88 | 70.46 | 77.34 | 73.52 | 69.04 | 62.26 | 65.50 | 66.22 | 57.58 | 44.78 | 68.62 | 56.64 | 48.26 | 53.72 | 64.46 |
| | | RDumb+CUI+CPAda | 78.68 | 70.14 | 69.36 | 76.94 | 71.97 | 66.62 | 60.17 | 63.16 | 63.46 | 53.04 | 40.24 | 67.70 | 54.70 | 46.28 | 51.84 | 62.29 |
| | | C-CoTTA+CUI | 76.84 | 74.02 | 71.80 | 76.20 | 74.22 | 66.34 | 57.00 | 56.16 | 61.28 | 49.22 | 40.34 | 65.32 | 49.68 | 43.80 | 44.12 | 60.42 |
| | | C-CoTTA+CUI+CPAda | 75.52 | 72.24 | 69.06 | 74.92 | 72.96 | 65.76 | 57.26 | 55.58 | 60.32 | 48.82 | 40.10 | 63.10 | 49.20 | 43.76 | 44.14 | 59.52 |
| | | RMT+CUI | 79.76 | 74.52 | 72.18 | 75.46 | 72.88 | 65.18 | 59.08 | 59.32 | 60.50 | 52.56 | 45.36 | 61.42 | 50.26 | 47.78 | 48.38 | 61.64 |
| | | RMT+CUI+CPAda | 77.19 | 71.89 | 69.69 | 72.71 | 70.69 | 62.97 | 57.25 | 56.83 | 58.75 | 51.33 | 43.75 | 58.99 | 49.09 | 46.21 | 47.37 | 59.65 |
| | 0.1 | Tent+CUI | 81.58 | 75.88 | 72.94 | 77.50 | 77.50 | 65.56 | 55.58 | 61.88 | 64.48 | 55.50 | 38.30 | 73.04 | 54.00 | 47.82 | 53.86 | 64.69 |
| | | Tent+CUI+CPAda | 81.38 | 74.74 | 72.92 | 77.14 | 74.02 | 65.06 | 55.80 | 61.90 | 62.88 | 51.48 | 38.02 | 71.96 | 51.04 | 47.50 | 53.22 | 62.60 |
| | | CoTTA+CUI | 87.14 | 85.42 | 84.06 | 85.26 | 84.26 | 74.72 | 66.20 | 67.42 | 68.04 | 55.16 | 42.72 | 75.58 | 55.92 | 49.34 | 54.18 | 69.03 |
| | | CoTTA+CUI+CPAda | 86.16 | 84.14 | 82.76 | 83.16 | 81.98 | 73.30 | 64.14 |  | 67.26 | 52.82 | 41.13 | 73.04 | 53.18 | 47.53 | 53.03 | 67.32 |
| | | SATA+CUI | 78.00 | 76.44 | 75.60 | 77.86 | 78.02 | 65.88 | 55.76 | 57.02 | 63.08 | 45.62 | 34.84 | 72.58 | 50.90 | 44.74 | 50.88 | 61.81 |
| | | SATA+CUI+CPAda | 76.16 | 75.34 | 74.32 | 77.32 | 76.06 | 65.24 | 54.34 | 56.50 | 63.36 | 45.00 | 34.96 | 70.48 | 50.22 | 43.96 | 49.82 | 60.87 |
| | | RDumb+CUI | 81.10 | 71.88 | 70.46 | 77.34 | 73.52 | 69.04 | 62.26 | 65.50 | 66.22 | 57.58 | 44.78 | 68.62 | 56.64 | 48.26 | 53.72 | 64.46 |
| | | RDumb+CUI+CPAda | 78.66 | 70.84 | 68.94 | 76.72 | 72.28 | 65.98 | 59.72 | 63.02 | 63.82 | 53.56 | 40.04 | 66.72 | 53.3 | 46.96 | 52.14 | 62.18 |
| | | C-CoTTA+CUI | 76.84 | 74.02 | 71.80 | 76.20 | 74.22 | 66.34 | 57.00 | 56.16 | 61.28 | 49.22 | 40.34 | 65.32 | 49.68 | 43.80 | 44.12 | 60.42 |
| | | C-CoTTA+CUI+CPAda | 75.52 | 72.38 | 69.58 | 75.76 | 73.16 | 65.28 | 56.52 | 55.28 | 60.96 | 49.36 | 39.68 | 64.04 | 48.86 | 42.68 | 43.90 | 59.53 |
| | | RMT+CUI | 79.76 | 74.52 | 72.18 | 75.46 | 72.88 | 65.18 | 59.08 | 59.32 | 60.50 | 52.56 | 45.36 | 61.42 | 50.26 | 47.78 | 48.38 | 61.64 |
| | | RMT+CUI+CPAda | 77.26 | 72.54 | 70.28 | 72.88 | 70.04 | 62.44 | 56.68 | 57.58 | 59.08 | 51.28 | 43.02 | 58.68 | 48.88 | 46.50 | 47.76 | 59.66 |
| | α | Method | Gau. | Sho. | Imp. | Def. | Gla. | Mot. | Zoo. | Sno. | Fro. | Fog | Bri. | Con. | Ela. | Pix. | Jpe. | Avg. |
| Efficiency First | 0.3 | Tent+CUI | 81.18 | 74.73 | 72.48 | 77.16 | 74.00 | 66.30 | 55.36 | 61.44 | 63.04 | 51.32 | 38.03 | 72.02 | 51.08 | 47.60 | 53.28 | 62.60 |
| | | Tent+CUI+CPAda | 81.04 | 73.22 | 72.42 | 76.97 | 65.32 | 65.79 | 54.33 | 61.31 | 62.01 | 50.28 | 37.02 | 71.04 | 50.58 | 46.90 | 54.28 | 61.50 |
| | | CoTTA+CUI | 81.38 | 74.78 | 73.00 | 77.24 | 74.10 | 66.20 | 55.68 | 61.72 | 63.20 | 51.30 | 38.08 | 71.90 | 50.94 | 47.62 | 53.36 | 62.70 |
| | | CoTTA+CUI+CPAda | 79.90 | 73.30 | 71.52 | 75.76 | 72.62 | 64.72 | 54.20 | 60.24 | 61.72 | 49.82 | 36.60 | 70.42 | 49.46 | 46.14 | 51.88 | 61.22 |
| | | SATA+CUI | 76.52 | 73.28 | 71.62 | 75.06 | 73.58 | 63.44 | 53.34 | 56.12 | 61.42 | 47.32 | 34.65 | 69.08 | 49.04 | 44.79 | 49.82 | 60.10 |
| | | SATA+CUI+CPAda | 74.93 | 71.69 | 70.03 | 73.47 | 71.99 | 61.85 | 51.75 | 57.05 | 59.83 | 45.73 | 33.06 | 67.49 | 47.45 | 43.20 | 48.23 | 58.52 |
| | | RDumb+CUI | 79.28 | 70.44 | 68.02 | 75.54 | 71.24 | 66.58 | 60.46 | 64.42 | 64.40 | 55.76 | 42.68 | 65.64 | 54.46 | 46.00 | 51.86 | 62.45 |
| | | RDumb+CUI+CPAda | 81.22 | 72.56 | 70.18 | 77.88 | 73.52 | 68.36 | 61.74 | 67.14 | 66.90 | 57.40 | 44.68 | 68.54 | 56.14 | 47.96 | 53.94 | 60.26 |
| | | C-CoTTA+CUI | 75.18 | 71.70 | 69.52 | 75.70 | 72.64 | 65.46 | 56.56 | 55.62 | 60.08 | 49.54 | 40.36 | 63.34 | 48.48 | 43.12 | 44.26 | 59.40 |
| | | C-CoTTA+CUI+CPAda | 74.09 | 70.56 | 68.41 | 74.59 | 71.53 | 64.35 | 55.53 | 54.51 | 58.97 | 48.23 | 39.15 | 62.24 | 47.37 | 42.21 | 43.65 | 58.36 |
| | | RMT+CUI | 77.84 | 72.06 | 69.98 | 73.04 | 71.28 | 64.52 | 56.94 | 57.16 | 59.54 | 51.18 | 43.22 | 59.16 | 48.46 | 45.70 | 46.54 | 59.80 |
| | | RMT+CUI+CPAda | 76.72 | 71.92 | 69.26 | 72.68 | 69.76 | 62.20 | 56.40 | 56.94 | 58.22 | 51.04 | 43.42 | 58.14 | 49.02 | 46.36 | 47.16 | 59.28 |
| | 0.2 | Tent+CUI | 81.18 | 74.73 | 72.48 | 77.16 | 74.00 | 66.30 | 55.36 | 61.44 | 63.04 | 51.32 | 38.03 | 72.02 | 51.08 | 47.60 | 53.28 | 62.60 |
| | | Tent+CUI+CPAda | 81.07 | 73.12 | 72.53 | 76.97 | 65.62 | 65.79 | 54.63 | 61.31 | 61.01 | 49.11 | 37.42 | 71.74 | 50.48 | 47.90 | 54.28 | 61.53 |
| | | CoTTA+CUI | 81.38 | 74.78 | 73.00 | 77.24 | 74.10 | 66.20 | 55.68 | 61.72 | 63.20 | 51.30 | 38.08 | 71.90 | 50.94 | 47.62 | 53.36 | 62.70 |
| | | CoTTA+CUI+CPAda | 79.98 | 73.38 | 71.60 | 75.84 | 72.70 | 64.80 | 54.28 | 60.32 | 61.80 | 49.90 | 36.68 | 70.50 | 49.54 | 46.22 | 51.96 | 61.30 |
| | | SATA+CUI | 76.52 | 73.28 | 71.62 | 75.06 | 73.58 | 63.44 | 53.34 | 58.64 | 61.42 | 47.32 | 34.65 | 69.08 | 49.04 | 44.79 | 49.82 | 60.10 |
| | | SATA+CUI+CPAda | 74.92 | 71.64 | 70.06 | 73.44 | 71.98 | 61.82 | 51.32 | 57.75 | 59.79 | 45.76 | 33.07 | 67.48 | 47.62 | 43.33 | 48.13 | 58.54 |
| | | RDumb+CUI | 79.28 | 70.44 | 68.02 | 75.54 | 71.24 | 66.58 | 60.46 | 64.42 | 64.40 | 55.76 | 42.68 | 65.64 | 54.46 | 46.00 | 51.86 | 62.45 |
| | | RDumb+CUI+CPAda | 76.64 | 67.88 | 66.06 | 73.14 | 69.18 | 64.14 | 58.36 | 62.18 | 62.56 | 53.62 | 40.90 | 64.56 | 52.20 | 43.90 | 49.62 | 60.32 |
| | | C-CoTTA+CUI | 75.18 | 71.70 | 69.52 | 75.70 | 72.64 | 65.46 | 56.56 | 55.62 | 60.08 | 49.54 | 40.36 | 63.34 | 48.48 | 43.12 | 44.26 | 59.40 |
| | | C-CoTTA+CUI+CPAda | 74.07 | 70.59 | 68.41 | 74.59 | 71.53 | 64.35 | 55.53 | 54.51 | 58.97 | 48.43 | 39.25 | 62.23 | 47.37 | 42.01 | 43.15 | 58.33 |
| | | RMT+CUI | 77.84 | 72.06 | 69.98 | 73.04 | 71.28 | 64.52 | 56.94 | 57.16 | 59.54 | 51.18 | 43.22 | 59.16 | 48.46 | 45.70 | 46.54 | 59.80 |
| | | RMT+CUI+CPAda | 76.68 | 72.54 | 70.06 | 72.36 | 70.56 | 62.40 | 57.28 | 56.64 | 59.16 | 51.36 | 43.26 | 58.70 | 48.86 | 46.08 | 47.18 | 59.25 |
| | 0.1 | Tent+CUI | 81.18 | 74.73 | 72.48 | 77.16 | 74.00 | 66.30 | 55.36 | 61.44 | 63.04 | 51.32 | 38.03 | 72.02 | 51.08 | 47.60 | 53.28 | 62.60 |
| | | Tent+CUI+CPAda | 81.07 | 73.12 | 72.53 | 76.97 | 65.62 | 65.79 | 54.63 | 61.31 | 61.01 | 49.11 | 37.42 | 71.74 | 50.48 | 48.90 | 54.48 | 61.60 |
| | | CoTTA+CUI | 81.38 | 74.78 | 73.00 | 77.24 | 74.10 | 66.20 | 55.68 | 61.72 | 63.20 | 51.30 | 38.08 | 71.90 | 50.94 | 47.62 | 53.36 | 62.70 |
| | | CoTTA+CUI+CPAda | 79.92 | 73.32 | 71.54 | 75.78 | 72.64 | 64.74 | 54.22 | 60.26 | 61.74 | 49.84 | 36.62 | 70.44 | 49.48 | 46.16 | 51.90 | 61.24 |
| | | SATA+CUI | 76.52 | 73.28 | 71.62 | 75.06 | 73.58 | 63.44 | 53.34 | 58.64 | 61.42 | 47.32 | 34.65 | 69.08 | 49.04 | 44.79 | 49.82 | 60.10 |
| | | SATA+CUI+CPAda | 74.94 | 71.61 | 70.02 | 73.46 | 72.07 | 61.93 | 51.31 | 57.86 | 59.80 | 45.72 | 33.62 | 67.43 | 47.62 | 43.81 | 48.58 | 58.65 |
| | | RDumb+CUI | 79.28 | 70.44 | 68.02 | 75.54 | 71.24 | 66.58 | 60.46 | 64.42 | 64.40 | 55.76 | 42.68 | 65.64 | 54.46 | 46.00 | 51.86 | 62.45 |
| | | RDumb+CUI+CPAda | 76.80 | 68.50 | 66.30 | 73.84 | 68.90 | 65.00 | 58.38 | 62.46 | 62.34 | 53.78 | 41.10 | 64.12 | 52.56 | 44.18 | 49.94 | 60.54 |
| | | C-CoTTA+CUI | 75.18 | 71.70 | 69.52 | 75.70 | 72.64 | 65.46 | 56.64 | 55.62 | 60.08 | 49.54 | 40.36 | 63.34 | 48.48 | 43.12 | 44.26 | 59.40 |
| | | C-CoTTA+CUI+CPAda | 74.09 | 70.76 | 68.41 | 74.59 | 71.53 | 64.35 | 55.53 | 54.44 | 58.94 | 48.23 | 39.14 | 62.23 | 47.37 | 42.63 | 43.65 | 58.39 |
| | | RMT+CUI | 77.84 | 72.06 | 69.98 | 73.04 | 71.28 | 64.52 | 56.94 | 57.16 | 59.54 | 51.18 | 43.22 | 59.16 | 48.46 | 45.70 | 46.54 | 59.80 |
| | | RMT+CUI+CPAda | 76.84 | 71.98 | 69.56 | 72.64 | 70.24 | 62.88 | 55.90 | 56.86 | 59.32 | 51.00 | 42.76 | 58.32 | 48.70 | 45.72 | 46.72 | 59.30 |

## D    LLM Usage Disclosure

We used LLMs solely to correct grammatical errors in the writing. The model was not involved in research design, data analysis, or result generation.

