# OpenReview forum: "Conformal Uncertainty Indicator for Continual Test-Time Adaptation"
_ICLR.cc/2026/Conference — ICLR 2026 Conference Withdrawn Submission_

### Official Review · Reviewer_C2fg · 2025-10-26

**Soundness:** 2
**Presentation:** 3
**Contribution:** 2
**Rating:** 4
**Confidence:** 3

**Summary:**

Continual Test-Time Adaptation (CTTA) aims to maintain model performance under sequential domain shifts during deployment. However, its heavy reliance on pseudo-labels often leads to error accumulation, underscoring the need for reliable uncertainty estimation. To tackle this challenge, the authors introduce the Conformal Uncertainty Indicator (CUI) — a plug-and-play framework that integrates Conformal Prediction (CP) with a small calibration buffer to provide statistically grounded uncertainty estimates. Extensive experiments across three benchmarks demonstrate that CUI delivers more calibrated uncertainty quantification and consistently enhances the robustness of diverse CTTA baselines.

**Strengths:**

1. The paper is clearly written, and the workflow diagrams effectively illustrate the proposed algorithm, making it easy for readers to follow the overall methodology.
2. The proposed approach consistently improves performance across multiple experimental settings and benchmark datasets.

**Weaknesses:**

1. Several prior TTA methods also mitigate the effect of unreliable supervision signals by down-weighting uncertain samples during loss computation, such as EATA [1] and DeYO [2]. Unlike the proposed method, these approaches do not require access to any source-domain data. Therefore, the motivation that incorporating source data is necessary to alleviate error accumulation is not entirely convincing.
2. The paper does not include direct performance comparisons with these representative reweighting methods (e.g., EATA [1], DeYO [2]), which would strengthen the empirical evaluation.
3. When comparing classification accuracy with the baselines, they did not utilize this part of the source-domain data, which makes the comparison somewhat unfair.
4. Some of the baselines used in the experiments are relatively outdated. It is recommended that the authors include more recent TTA approaches for more comprehensive comparison.
5. In Figure 4, the names of the baselines are concatenated or even overlap in places, making the figure difficult to read. The authors are encouraged to increase the spacing between labels to improve clarity and presentation quality.

[1] Efficient Test-Time Model Adaptation without Forgetting. ICML'22

[2] Entropy is not Enough for Test-Time Adaptation: From the Perspective of Disentangled Factors. ICLR'24

**Questions:**

See weaknesses.

---

### Official Review · Reviewer_sfXD · 2025-10-26

**Soundness:** 3
**Presentation:** 3
**Contribution:** 3
**Rating:** 6
**Confidence:** 4

**Summary:**

The paper proposes a Conformal Uncertainty Indicator (CUI) for continual test-time adaptation. CUI estimates joint model+data shift via a JS-divergence score, inflates the conformal quantile accordingly, and then forms a prediction set by including all classes whose non-conformity scores fall below the compensated threshold; the set size serves as a reliability signal that also weights adaptation updates to reduce error accumulation under shift.

**Strengths:**

1. The paper is clearly written, with a step-by-step presentation of the conformal construction and how it interfaces with TTA.

1. The paper tackles a timely and important problem: calibration-aware test-time adaptation, aiming to deliver coverage guarantees alongside accuracy.

1. The method is model-agnostic at inference and could, in principle, be layered on top of diverse TTA strategies.

**Weaknesses:**

Major weaknesses

1. Missing related work on calibration during adaptation, TEA (Test-time Energy Adaptation) [1], which directly adapts a model via energy minimization and reports effects on calibration.

1. Missing related work on TTA accuracy/uncertainty estimation, AETTA [2], which provides label-free accuracy estimation during TTA. Positioning against AETTA’s reliability signals is essential.

1. The paper leverages standard split-conformal machinery with a shift correction term; please clarify whether the correction has any theoretical coverage guarantee (even approximate) or is purely heuristic.

1. The paper does not report calibration metrics (ECE/MCE) alongside accuracy to isolate calibration gains from accuracy gains.


Minor weaknesses

1. Figure 1 and its caption could be improved by explaining $\gamma$ and $L(x)$.






[1] Yuan, Yige, et al. "TEA: Test-time energy adaptation." Proceedings of the IEEE/CVF Conference on Computer Vision and Pattern Recognition. 2024.

[2] Lee, Taeckyung, et al. "AETTA: Label-free accuracy estimation for test-time adaptation." Proceedings of the IEEE/CVF Conference on Computer Vision and Pattern Recognition. 2024.

**Questions:**

1. The method assumes a small calibration set from the source domain. The setting makes sense, but might limit the applicability. How would the method work if we utilize a few test set data (with breaking the domain shift estimation)?


1. Is the shift-compensation term theoretically motivated or purely empirical?

1. As the model adapts, the score distribution drifts. Do you re-estimate the conformal quantile periodically? If not, how do you maintain coverage over time?

1. Please include sensitivity to calibration set size, choice of non-conformity score (confidence vs. energy vs. margin), and shift-metric (JS vs. other divergences).

---

### Official Review · Reviewer_oQ5d · 2025-10-28

**Soundness:** 3
**Presentation:** 2
**Contribution:** 2
**Rating:** 4
**Confidence:** 5

**Summary:**

in continual test-time adaptation, models adapt via pseudo-labels(or in a similar manner), making reliable uncertainty estimation critical. The paper proposes a relaxed setting where a static calibration buffer of labeled source-domain samples are available, and builds their method on top. The main methodology presented is CUI - it measures both the model's divergence from source and how much the sample-level predictions diverge.

**Strengths:**

* Easy to follow
* Proposes an efficient yet effective solution, which can be integrated on-top of existing TTA methods.

**Weaknesses:**

Comparison against recent baselines: https://openreview.net/pdf?id=USWkUOfxOO
* This is a method that uses a simple technique for calibration, with strong empirical performance. I wish the authors could compare their uncertainty estimation module against this. Furthermore, PseudoCal can also be used to create balanced samples for test-time adaptation WITHOUT additional source sample memory.
Questions against spurious correlations

Comparison against datasets of (https://arxiv.org/abs/2403.07366)
* Are the calibrations (uncertainty measures) reliable against spurious correlations as well? There are works that point out that upon spurious corrleations (such as waterbirds datasets) undermine the performance of such methods, and I wish the authors could identify their methods gains against DeYO, on the waterbirds dataset.

For comparison against methods that utilize replay buffer (https://arxiv.org/abs/2208.05117, https://arxiv.org/abs/2310.10074)
* There are multiple methods that utilize a memory buffer(reliable) upon noisy / domain-shift schemes.
* The proposed method of authors utilize a g.t. source label during TTA, thus should perform almost always better on all test scenarios, than methods which keep additional sets of memory sampled from the test batch.

Ablations
* It is well-known that upon multiple adaptations, the model slowly degrades in performance due to error accumulation in TTA.
* It would be meaningful, if the authors could provide an ablation, by showing 1) upper-bound (i.e. update with test labels) 2) lower-bound (i.e. vanilla TENT-style adaptation) 3) reset-style TTA (reset the model every batch) and 4) proposed method. An ideal scenario would be that the proposed method significantly recovers 1) while significantly pushing the point where 2) degrades below 3).

I'm willing to raise my score if the ablative results are supportive of the authors' arguments.

**Questions:**

Refer to above

---

### Official Review · Reviewer_hXeF · 2025-11-03

**Soundness:** 2
**Presentation:** 3
**Contribution:** 1
**Rating:** 2
**Confidence:** 3

**Summary:**

This paper suggests the reliable uncertainty metric to adapt the models in test time only with the confident and likely correct data, preventing error accumulation caused by adapting models with false pseudo-labels. To this end, the authors propose the conformal uncertainty indicator that can compensate the gaps induced by the distribution shifts, by assuming that a small number of source data is available during test time. The proposed method achieves the improved performances in continual domain shifts compared to baselines, and show easy-to-use applications to various existing test-time adaptation (TTA) methods.

**Strengths:**

- This paper presents the uncertainty estimator for how reliable the predicted test data is with theoretical backgrounds.
- The proposed method, CUI, can be equipped with different TTA methods with reasonable amount of complexity in computation and resources.

**Weaknesses:**

- I’m still convinced how crucial or urgent the main problem this paper is motivated to tackle with — the error accumulation induced by updating with unreliable pseudo-labels. When comparing the OOD accuracy of the baseline that might suffer the error accumulation and those with CUI, the overall error gaps are marginal (e.g., less than 2% in most of the benchmarking cases in Table 1 and authors did not even report the error bars). When considering that the baselines do not involve any additional conditions, such as availability of source data or models, TTA with CUI does not present the interesting trade-off between decreasing robustness against OOD versus additional conditions w.r.t. computations or privacy issues. I believe the proposed methods would be much more appreciated in scenarios where the unreliable pseudo-labels are critically undermining the robustness, and CUI can substantially increase the robustness with reasonable conditions.
- In Figure 4, how could the additional memory usage be small with inferencing with additional source model for calibration data inferencing?

**Questions:**

Please see weaknesses.

---

### Note · Authors · 2025-11-28

I have read and agree with the venue's withdrawal policy on behalf of myself and my co-authors.